# Cell ontology guided transcriptome foundation model

**Xinyu Yuan[1,2], Zhihao Zhan[1,2], Zuobai Zhang[1,2], Manqi Zhou[4]**
**Jianan Zhao[1,2], Boyu Han[3], Yue Li[1,3,*], Jian Tang[1,5,6,*]**
[1]Mila - Québec AI Institute, [2]University of Montréal
[3]McGill University, [4]Cornell University, [5]HEC Montréal, [6]CIFAR AI Chair
[*]Correspondence: liyue@mila.quebec; tangjian@mila.quebec

## Abstract

Transcriptome foundation models (TFMs) hold great promises of deciphering the transcriptomic language that dictate diverse cell functions by self-supervised learning on large-scale single-cell gene expression data, and ultimately unraveling the complex mechanisms of human diseases. However, current TFMs treat cells as independent samples and ignore the taxonomic relationships between cell types, which are available in cell ontology graphs. We argue that effectively leveraging this ontology information during the TFM pre-training can improve learning biologically meaningful gene co-expression patterns while preserving TFM as a general purpose foundation model for downstream zero-shot and fine-tuning tasks. To this end, we present **s**ingle **c**ell, **C**ell-**o**ntology guided TFM (scCello). We introduce cell-type coherence loss and ontology alignment loss, which are minimized along with the masked gene expression prediction loss during the pre-training. The novel loss component guide scCello to learn the cell-type-specific representation and the structural relation between cell types from the cell ontology graph, respectively. We pre-trained scCello on 22 million cells from CellxGene database leveraging their cell-type labels mapped to the cell ontology graph from Open Biological and Biomedical Ontology Foundry. Our TFM demonstrates competitive generalization and transferability performance over the existing TFMs on biologically important tasks including identifying novel cell types of unseen cells, prediction of cell-type-specific marker genes, and cancer drug responses. Source code and model weights are available at https://github.com/DeepGraphLearning/scCello.

## 1   Introduction

Cells are basic units of all living organisms. Deciphering diverse cell functions through gene expression is a long-standing challenge in life science and yet the essential path towards precision and personalized medicine. In this context, single-cell RNA sequencing (scRNA-seq) has emerged as a pivotal technique to measure the gene expression in individual cells. The vast amount of publicly available scRNA-seq data offers a rich transcriptomic data source [48] for learning cell representations towards various research applications, such as cancer therapy [62] and drug discovery [5].

Recently, several *Transcriptome Foundation Models* (TFMs) were developed to improve cell representation learning. They mainly utilize pre-training methods analogous to natural language processing like masked token prediction, treating genes as "tokens" and cells as "sentences" [14, 61, 70, 55]. However, the existing TFMs treat cells as independent samples and ignore their cell-type lineages. On the other hand, prior knowledge of the taxonomic relationships of cell types has been made available through the cell ontology graph by Open Biological and Biomedical Ontology Foundry [4]. Effectively leveraging the ontology knowledge can improve the quality of the pre-training on large-scale scRNA-seq atlases, which are heterogeneous and encompass hundreds of cell types. This can be done by training the TFM to recognize the inherent ontology relationships among cell types, thereby

refining the cell representations. For instance, "mature $\alpha$-$\beta$ T cell" should be closer to "mature T cells" compared to more general term "T cells" and farther from neurons and astrocytes from the brain (e.g., Tab. 7).

To capture this intuition, we propose scCello, a **s**ingle **c**ell, **Cell-o**ntology guided TFM. scCello learns cell representation by integrating cell type information and cellular ontology relationships into its pre-training framework. scCello's pre-training framework is structured with three levels of objectives: (1) **gene level**: a masked token prediction loss to learn gene co-expression patterns, enriching the understanding of gene interactions (Sec. 2.2); (2) **intra-cellular level**: an ontology-based cell-type coherence loss to encourage cell representations of the same cell type to aggregate, prompting consistency between cells and their types (Sec. 2.3); and (3) **inter-cellular level**: a relational alignment loss to guide the cell representation learning by consulting the cell-type lineage from the cell ontology graph (Sec. 2.4)..

We demonstrate the generalizability and transferability of scCello on 22 million cells from CellxGene. For model generalization, we observe that scCello excels on cell type identification across all datasets in both zero-shot setting (i.e., directly using the pre-trained model) (Sec. 4.2.1) and fine-tuning setting (Sec. 4.2.2). In particular, scCello accurately classifies novel cell types by leveraging the ontology graph structure (Sec. 4.3). For transferability, scCello demonstrates competitive performances in predicting cell-type-specific marker genes (Sec. 4.4) and cancer drug responses (Sec. 4.5). Additionally, scCello is robust against batch effects (Sec. 4.6). Finally, we validate our contribution via ablation study (Sec. 4.7).

## 2 Method

Fig. 1 illustrates an overview of scCello. We present the details of individual components below.

### 2.1 Data Preprocessing

**Cell ontology graph.** Cell ontology is a widely used metadata schema for standard cell type annotations [16]. We downloaded the ontology from Open Biological and Biomedical Ontology Foundry (`https://obofoundry.org/`). It is structured as an unweighted directed acyclic graph $\mathcal{G} = (\mathcal{V}, \mathcal{E})$, where each node $v \in \mathcal{V}$ corresponds to a distinct cell type and each directed edge $(u, v) \in \mathcal{E}$ denotes a hierarchical lineage relationship of the form "is a subtype of" between cell types (Fig. 1a). To accurately represent the inherently symmetric "being biologically similar" relationship between cell types, the directed graph was transformed into an undirected one for subsequent calculation of cellular ontology relationships in Sec. 2.4.

**scRNA-seq data.** The scRNA-seq data were downloaded from CellxGene. After the preprocessing (App. B), we obtained 22 million cells. Each single-cell transcriptome is represented by a sequence of tuples, each containing genes and their expression counts.[1] Each sequence was then ordered by the rank of the gene expression values [61] , akin to the sequential ordering of natural languages. Given a batch of $B$ cells, each cell $i \in \{1, \ldots, B\}$ was assigned a cell type ontology identifier $c_i \in \mathcal{V}$ from the CellxGene database, to enable mapping between cell and cell ontology.

### 2.2 Masked Gene Prediction

Same as BERT [15], scCello predicts a randomly masked gene token in each cell based on its surrounding context in the sequence. This objective $\mathcal{L}_{\text{MGP}}$ aims to learn the dynamic gene co-expression network.

### 2.3 Intra-Cellular Ontology Coherence

A straightforward approach to encourage learning the cell representations that are coherent to the cell type labels is to apply cross-entropy loss for supervised cell type classification. However, this approach is limited in learning cell representation for the foundation model. Instead, we employed a supervised contrastive loss as our objective $\mathcal{L}_{\text{Intra}}$, which directly optimizes the TFM rather than

---

[1]scRNA-seq data was from CellxGene database `https://cellxgene.cziscience.com/`.

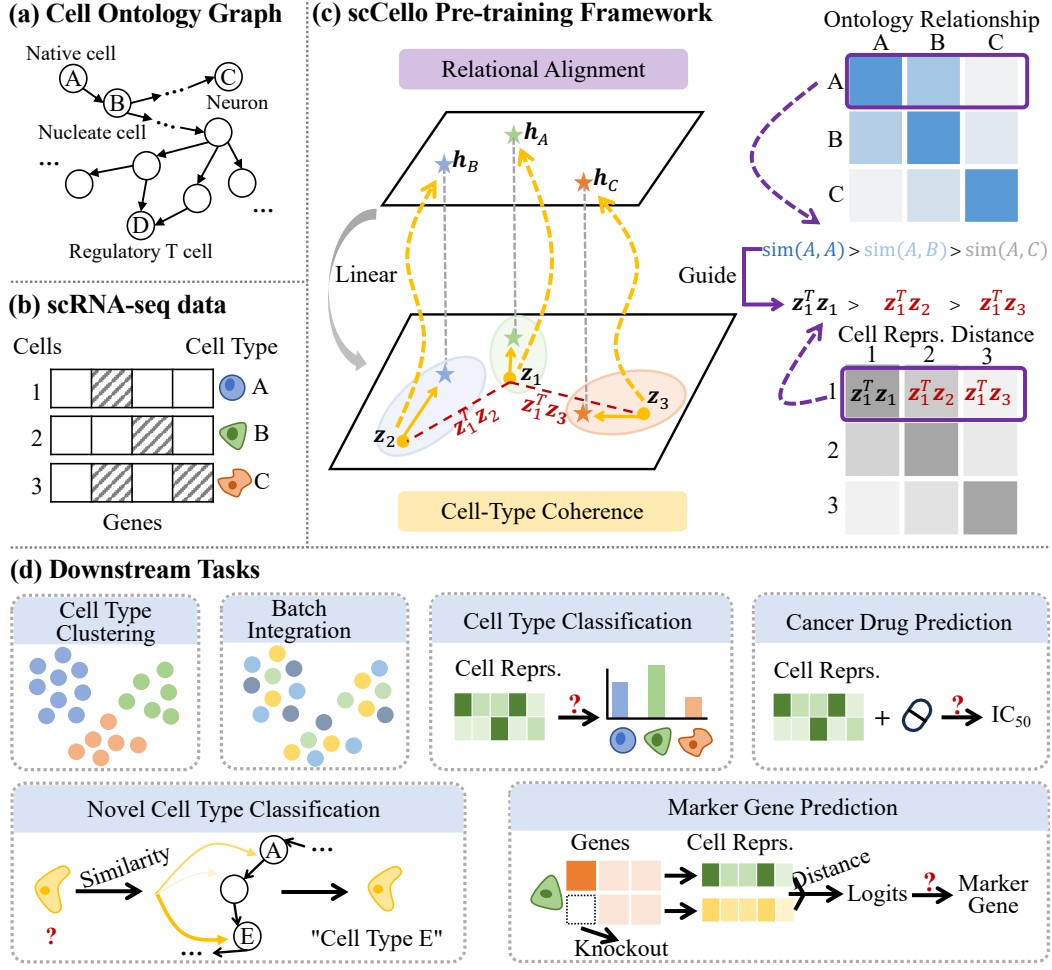

Figure 1: (a) Cell ontology graph describes taxonomic relationships between cell types. (b) Each cell in scRNA-seq data is represented by gene sequences, and associated with a cell type ontology identifier. (c) The pre-training framework of scCello is structured with three levels of objectives: gene-level masked gene prediction, intra-cellular level cell type coherence and inter-cellular level ontology alignment. For example, as shown in panel b, cells 1, 2, and 3 are labelled with cell type A, B and C. The intra-cellular cell type coherence loss encourages alignment of embedding $\mathbf{z}_1$ with $\mathbf{h}_A$, $\mathbf{z}_2$ with $\mathbf{h}_B$, and $\mathbf{z}_3$ with $\mathbf{h}_C$. The inter-cellular level ontology alignment loss encourages representational learning of cell similarities $\mathbf{z}_i^\top \mathbf{z}_j$ between cell $i$ and $j$ to be consistent to the similarity of their corresponding cell types $sim(c_i, c_j)$ based on the ontology relationships. (d) Downstream tasks enabled by scCello and demonstrated in the study.

merely learning through the linear classifier:

$$\mathcal{L}_{\text{Intra}} = -\sum_{i=1}^{B} \log \left( \frac{\exp(\boldsymbol{z}_i^T \boldsymbol{h}_{c_i}/\tau)}{\exp(\boldsymbol{z}_i^T \boldsymbol{h}_{c_i}/\tau) + \sum_{j=1, j \neq i}^{B} \exp(\boldsymbol{z}_i^T \boldsymbol{h}_{c_j}/\tau)} \right). \tag{1}$$

where $\boldsymbol{z}_i$ and $\boldsymbol{h}_{c_i}$ denote the latent representation of cell $i$ and cell type $c_i$, respectively.

This supervised contrastive loss pulls representations of the same class (positives) and repels representations of different classes (negatives). It often leads to representations that are at least as discriminative as the cross-entropy loss [22]. To reduce the degrees of freedom available for TFM optimization, we introduce a regularization term $\mathcal{L}_{\text{Reg}}$:

$$\mathcal{L}_{\text{Reg}} = \sum_{i=1}^{B} \|\text{Linear}(\boldsymbol{h}_{c_i}) - \boldsymbol{z}_i\|_2^2, \tag{2}$$

where the linear layer is shared across all cells and cell types. Thereby, it constrains the cell type representation space to be an affine transformation of the cell representation space.

## 2.4 Inter-Cellular Relational Alignment

To encourage TFMs to learn inter-cellular ontology relationships, scCello forces cell representations to truthfully reflect the pairwise node structural similarity derived from the cell ontology graph, using a relational alignment objective. This objective constitutes the most important part of scCello.

**Ontology relationships.** To effectively quantify ontology relationships between cell types from the ontology graph, scCello estimates pairwise node structural similarities as proxies using Personalized PageRank (PPR) [20]. PPR is a graph learning algorithm. The PPR score $\text{PPR}(u, v)$ estimates the probability for a random walk. It starts from a given target node $u \in \mathcal{V}$ and terminates at another node $v \in \mathcal{V}$. Importantly, this is a context-sensitive structural similarity measure that accounts both direct connections and broader subgraph patterns [67]. It also provides robustness against variations in global network structures, such as variable node degrees and clustering coefficients [11]. To improve robustness (as justified in App. A), we transform $\text{PPR}(\cdot)$ through a non-linear function to derive the structural similarities $\text{sim}(\cdot)$ as ontology relationships tunable by a hyper-parameter threshold $s$:

$$\text{sim}(u, v) = \begin{cases} \lfloor \log_2(\frac{\text{PPR}(u,v)}{s} + 1) \rfloor, & \text{if } \text{PPR}(u, v) \geq s \\ 1, & \text{otherwise} \end{cases}. \tag{3}$$

**Relational alignment.** Cells with closely related cell types tend to be more similar than those with distinct cell types. This observation guides scCello to align the distances between cell representations *w.r.t.* a target cell, with their structural similarities $\text{sim}(\cdot)$ (as shown in Fig. 1c). Specifically, given a batch of $B$ cells, if we consider a target cell $i$ and another cell in the batch $j \neq i$, the representation distance $\boldsymbol{z}_i^T \boldsymbol{z}_j$ should reflect their structural similarity $\text{sim}(c_i, c_j)$. Accordingly, a negative sample set $\Omega_{i,j} = \{k | \text{sim}(c_i, c_j) > \text{sim}(c_i, c_k), 1 \leq k \leq B\}$ can be produced, where cell pair $(i, k)$ are considered less similar to the cell pair $(i, j)$ and should be contrasted against in the representation space using the objective $\mathcal{L}_{\text{Inter}}$:

$$\mathcal{L}_{\text{Inter}} = -\sum_{i=1}^{B} \sum_{j=1, j \neq i}^{B} \log \left( \frac{\exp(\boldsymbol{z}_i^T \boldsymbol{z}_j / \tau)}{\exp(\boldsymbol{z}_i^T \boldsymbol{z}_j / \tau) + \sum_{k \in \Omega_{i,j}} \exp(\boldsymbol{z}_i^T \boldsymbol{z}_k / \tau)} \right). \tag{4}$$

Notably, ancestor cell types, which can reach the target cell type via the directed "is a subtype of" edge on the ontology graph, are structurally distant from the target cell type. Despite being distant, they fall into the same, broader cell type category. Contrasting cells associated with these distant ancestor cell types with the target cell is counter-intuitive. Therefore, scCello explicitly excludes such cells from the negative sample set, avoiding inappropriately pushing away biologically similar cells. This enhances scCello's capability to discern subtle similarities and differences within the cell types.

## 2.5 Overall Pre-training Objective

During pre-training, we seek to minimize the loss functions of all pre-training tasks simultaneously:

$$\theta^* \leftarrow \arg\min_{\theta} \ \mathcal{L}_{\text{MGP}} + \mathcal{L}_{\text{Inter}} + \mathcal{L}_{\text{Intra}} + \mathcal{L}_{\text{Reg}} \tag{5}$$

where $\theta$ denotes all learnable parameters in scCello, which adopts transformer stacks as model backbones. We state the detailed information of model architectures in App. D.

## 3 Related Work

The rapid growth of scRNA-seq datasets has opened new avenues for constructing TFMs, enabling transfer learning across various biological downstream tasks. Initial efforts, such as scBERT [73], Exceiver [13] and Geneformer [61], borrows the concept of masked language modeling [15] from natural language processing (NLP) domain for pre-training, by treating cells as sentences and genes as tokens. Concurrently, tGPT [59] and scGPT [14] explored generative modeling [53], and CellLM [76]

adapted the idea of contrastive learning [42]. Following the concept of "scaling" towards emergent ability [70] in NLP, scFoundation [25] proposes the largest foundation model at the time in terms of model size and pre-training data size; scHyena [49] scales modeling context window size to the full length of scRNA-seq data with Hyena operator [51] instead of conventionally used transformers. scTab [18] is the first to explore large-scale supervised learning mechanism for scRNA-seq pre-training, and is capable of annotating unseen tissue cells for real-world applications. Moreover, SCimilarity [28] and UCE [55] focus on developing a unified latent space as a large-scale reference atlas for querying new cells. Yet, these TFMs mainly treat cells as independent samples during training and ignore their biological ontology relationships. scCello bridges this gap by incorporating cell type relationships derived from the cell ontology graph into TFM pre-training. This strengthens TFMs' model generalization and transferability capability, as shown in Sec. 4.

# 4 Experiments

As an overview, the following experiments show that, **(1)** scCello can generalize to unseen cells, and to more difficult settings, such as cells of unseen cell types, tissues, and donors (Sec. 4.2.1); **(2)** scCello can benefit from fine-tuning on target datasets (Sec. 4.2.2); **(3)** the structural similarity embedded in scCello helps to classify novel cell types in a zero-shot manner (Sec. 4.3); **(4)** scCello effectively transfers to different downstream tasks (Sec. 4.4 and Sec. 4.5); **(5)** scCello is robust to batch effects that arise from different experimental conditions (Sec. 4.6); **(6)** Each loss component in Eqn. 5 is beneficial to scCello (Sec. 4.7). For every table reported, we used **bold** to highlight the best performance and results within 0.005 difference from the best. We used underlining to denote the second-best performances. For all metrics, ↑ indicates the higher the better.

## 4.1 Setups

**Pre-training and downstream datasets.** We collected a large pre-training dataset consisting of 22 million cells along with downstream datasets. In particular, we generated one in-distribution (ID) and six out-of-distribution (OOD) datasets (App. B). The ID dataset is denoted as $D^{id}$. For the OOD setting, we introduced three scenarios: unseen **c**ell **t**ypes ($\{D_i^{ct}\}_{i=1}^2$), unseen cell **tis**sues ($\{D_i^{ts}\}_{i=1}^2$), and unseen **do**nors ($\{D_i^{dn}\}_{i=1}^2$). Each scenario has two datasets. Notably, the OOD donor setting presents more realistic challenges than ID and other OOD settings because of the potential batch effects in the test donors.

**Pre-training configurations.** An Adam optimizer [38] (learning rate: 0.001, weight decay: 0.001, warm-up steps: 3,333) was used to train the scCello for 40,000 steps on 4 NVIDIA A100 GPUs on Compute Canada. We used 192 for batch size. More details are introduced in App. D.

**Baselines.** Across all downstream tasks, scCello is benchmarked with leading open-source large-scale TFMs: Geneformer [61], scGPT [14], scTab [18], UCE [55], and three TFM ablations. We also implemented ablated versions of scCello that only differ in the pre-training objectives from scCello: scCello using only the masked gene prediction loss (denoted as MGP), scCello using only the cell type supervised classification (denoted as Sup), and scCello using only the two losses (denoted as MGP+Sup). The three ablated TFMs provide a reference to isolate the effect of implementation details and training configurations. For each task, we also selected state-of-the-art non-TFM methods for fair comparison.

**Downstream metrics.** We evaluated the 3 tasks by the following metrics. (1) Clustering metrics include normalized mutual information (NMI), adjusted rand index (ARI), average silhouette width (ASW), and the average of the 3 scores (AvgBio) to assess both between-cluster separation and within-cluster closeness [14]. The batch integration task (Sec. 4.6) is evaluated by $\text{ASW}_b$, graph connectivity (GraphConn) and their average (AvgBatch), along with an overall score (Overall = $0.6 \times \text{AvgBio} + 0.4 \times \text{AvgBatch}$) to balance biological relevance and batch consistency following [14]. (2) Classification metrics include accuracy (Acc), Macro F1 and area under the ROC curve (AU-ROC) [50]. (3) Regression task metrics include Pearson correlation coefficient score (PCC) [50]. Details for each metric were provided in App. E.1.

Table 1: Zero-shot cell type clustering on the curated ID and OOD datasets.

| Method | In-Distribution (ID) | | | | Out-of-Distribution (OOD) | | | | | | |
| --- | --- | --- | --- | --- | --- | --- | --- | --- | --- | --- | --- |
| | $D^{id}$ | | | | $D_1^{ct}$ | $D_2^{ct}$ | $D_1^{ts}$ | $D_2^{ts}$ | $D_1^{dn}$ | $D_2^{dn}$ | |
| | NMI↑ | ARI↑ | ASW↑ | AvgBio↑ | AvgBio↑ | AvgBio↑ | AvgBio↑ | AvgBio↑ | AvgBio↑ | AvgBio↑ | OOD Avg.↑ |
| **Non-TFM Methods** | | | | | | | | | | | |
| Raw Data | 0.566 | 0.237 | 0.453 | 0.419 | 0.703 | 0.629 | 0.540 | 0.631 | 0.458 | 0.460 | 0.570 |
| Seurat | 0.648 | 0.270 | 0.407 | 0.442 | 0.752 | 0.737 | 0.587 | 0.636 | 0.466 | 0.489 | 0.611 |
| Harmony[1] | 0.621 | 0.261 | 0.382 | 0.421 | 0.432 | 0.417 | 0.462 | 0.515 | 0.456 | 0.474 | 0.459 |
| scVI | 0.660 | 0.297 | 0.464 | 0.474 | 0.760 | 0.725 | 0.577 | 0.634 | 0.478 | 0.502 | 0.613 |
| **Ontology-Agnostic TFMs** | | | | | | | | | | | |
| Geneformer | 0.616 | 0.261 | 0.418 | 0.432 | 0.689 | 0.668 | 0.539 | 0.597 | 0.468 | 0.482 | 0.574 |
| scGPT | 0.615 | 0.258 | 0.442 | 0.438 | 0.707 | 0.720 | 0.544 | 0.627 | 0.456 | 0.477 | 0.589 |
| scTab | 0.707 | 0.479 | 0.544 | 0.577 | 0.759 | 0.726 | 0.515 | 0.657 | OOM | OOM | / |
| UCE | 0.670 | 0.304 | 0.494 | 0.489 | **0.772** | 0.741 | 0.598 | 0.670 | 0.485 | 0.506 | 0.629 |
| MGP | 0.662 | 0.306 | 0.451 | 0.473 | 0.714 | 0.740 | 0.576 | 0.628 | 0.488 | 0.518 | 0.611 |
| Sup | 0.703 | 0.393 | 0.569 | 0.555 | **0.767** | 0.775 | 0.605 | 0.680 | 0.552 | 0.573 | 0.659 |
| MGP+Sup | 0.661 | 0.337 | 0.550 | 0.516 | 0.758 | 0.764 | **0.610** | 0.672 | 0.553 | 0.570 | 0.655 |
| **Ontology-Enhanced TFMs** | | | | | | | | | | | |
| **scCello** | **0.785** | **0.558** | **0.667** | **0.670** | 0.769 | **0.786** | **0.612** | **0.705** | **0.608** | **0.643** | **0.687** |

[1] Harmony could be over-corrected *w.r.t.* batch labels for datasets with many batches [10].

## 4.2 Cell Type Identification

### 4.2.1 Zero-shot Cell Clustering Results

**Setup.** For the cell type clustering task, TFM baselines and four non-TFM methods were evaluated: (1) raw data expressions of highly variable genes (*abbr.*, Raw Data) [34]; (2) Seurat [27]; (3) Harmony [40] (4) scVI [44]. Cell representations were extracted from the baselines and clustered by Louvain algorithm [6]. We evaluated the clustering performance of each method on both ID dataset $D^{id}$ and OOD datasets $D_i^{cond}$ ($cond \in \{ct, ts, dn\}$, $i \in \{1, 2\}$).

**ID and OOD generalization.** We reported zero-shot cell type clustering performance in Tab. 1, and included all the metrics for all datasets in App. E.2.1 due to space constraint. For both the ID and OOD settings, scCello consistently outperforms all baselines, achieving a 16.1% improvement in AvgBio on the ID dataset and a 12.1% improvement in average AvgBio across the six OOD datasets. Interestingly, while scCello outperforms non-TFM methods by a large margin, Geneformers and scGPT barely surpass these methods. The latter is consistent with previous observations [75].

In the OOD experiments, scCello confers strong generalization capability across unseen cell types tissue, and donors. In cell type clustering, scCello is the second best only trailing UCE by 0.03 and the best method for dataset 1 and 2. The OOD tissue setting highlights scCello's ability to transfer its learned knowledge to different unseen tissues. Specifically, scCello achieve 0.6 and 0.7 while most methods conferred below 0.6 and 0.7 for the two datasets, respectively. For the unseen OOD donor scenario, most methods perform poorly with AvgBio ranging between 0.45 and 0.55. scCello led the chart achieving AvgBio above 0.6 in both datasets. Overall, scCello showcases strong model generalization capabilities across a range of biological conditions, which is attributable to the integration of cell ontology priors during its TFM pre-training. Indeed, the ablated models namely MGP, Sup, and MGP+Sup conferred lower scores compared to the full model.

### 4.2.2 Fine-tuning Results

**Setup.** We benchmarked all TFM baselines except UCE for its lack of fine-tuning support. These TFMs were fine-tuned on a subset of our pre-training data with supervised classification loss (details in App. E.2.2). We assessed both classification and clustering performance on the ID dataset $D^{id}$. We also compared with a non-TFM method, scANVI [72].

**Improvement with fine-tuning.** In Tab. 2, The fine-tuned scCello outperforms other TFMs and non-TFM methods on both classification and clustering metrics, achieving up to 25.9% improvement in Macro F1 over the best baseline. Moreover, scCello without fine-tuning still surpasses the performance of the other fine-tuned methods, further highlighting its superior transferability.

Table 2: Cell type identification using fine-tuned TFMs. Both the classification and clustering performances on the ID dataset $D^{id}$ are reported.

| Method | Classification | | Clustering |
|---|---|---|---|
| | Acc ↑ | Macro F1 ↑ | AvgBio ↑ |
| **Non-TFM Methods** | | | |
| scANVI | 0.763 | 0.490 | 0.568 |
| Scratch | 0.621 | 0.223 | 0.544 |
| **Ontology-Agnostic TFMs** | | | |
| Geneformer | 0.747 | 0.440 | 0.439 |
| scGPT | 0.712 | 0.344 | 0.477 |
| scTab | 0.778 | 0.373 | 0.606 |
| MGP | 0.722 | 0.287 | 0.607 |
| Sup | 0.812 | 0.363 | 0.659 |
| MGP+Sup | 0.820 | 0.406 | 0.607 |
| **Ontology-Enhanced TFMs** | | | |
| **scCello** | **0.867** | **0.511** | **0.694** |

Table 3: Marker gene prediction, a binary classification task to identify cell-type-specific marker genes.

| Method | $D_1^{mk}$ | $D_2^{mk}$ | Avg.↑ |
|---|---|---|---|
| | AUROC ↑ | AUROC ↑ | |
| **Non-TFM Methods** | | | |
| DET | 0.721 | 0.683 | 0.702 |
| **Ontology-Agnostic TFMs** | | | |
| Geneformer | 0.452 | 0.470 | 0.461 |
| scGPT | 0.385 | 0.387 | 0.386 |
| scTab | 0.672 | 0.727 | 0.700 |
| UCE | 0.500 | 0.500 | 0.500 |
| MGP | 0.579 | 0.629 | 0.604 |
| Sup | 0.699 | 0.693 | 0.696 |
| MGP+Sup | 0.730 | **0.730** | 0.730 |
| **Ontology-Enhanced TFMs** | | | |
| **scCello** | **0.756** | **0.729** | **0.743** |

Table 4: Cancer drug response prediction: a regression task to predict the $IC_{50}$ values of drugs.

| Method | Non-TFM Methods | Ontology-Agnostic TFMs | | | | | | | | Ontology-Enhanced TFMs |
|---|---|---|---|---|---|---|---|---|---|---|
| | DeepCDR | scFoundation | Geneformer | scGPT | scTab | UCE | MGP | Sup | MGP+Sup | scCello |
| PCC ↑ | 0.854 | 0.882 | 0.911 | **0.919** | 0.913 | **0.922** | 0.872 | 0.915 | 0.916 | **0.917** |

## 4.3 Novel Cell Type Classification

Novel cell type classification aims to label cells of unseen cell types without further fine-tuning. This task is useful for annotating completely new scRNA-seq datasets but infeasible for most of the supervised methods that solely rely on the labels observed in the training data [8, 31, 68]. Leveraging the cell ontology graph that comprises the lineage relations among all of the known cell types, scCello makes this task feasible.

**Setup.** Our goal is to classify new query cells into "novel cell types" not seen during pre-training. To do this, we generate representations for both query cells and novel cell types, using similarity measures for classification. This process involves utilizing similarities between TFM-derived representations for the former and biological relationships from the cell ontology graph for the later. Details were described in App. E.3.

We benchmarked all TFMs and evaluated them on OOD cell type datasets $D_1^{ct}$ and $D_2^{ct}$. We increased the difficulty of this task by the number of novel cell types (#Cell Types) that exist among the query cells. Specifically, we simulated five difficulty levels, with the number of novel cell types ranging from 10% to 100% of the total cell types. To assess the variance of the performance, we randomly sampled cell type combinations 20 times at each level.

**OOD generalization.** In Fig. 2, scCello led other TFMs by a large margin, achieving up to 76.8% Acc to classify 9 novel cell types (i.e., 10% of the total heldout cell types) and 33.5% Acc to classify up to 87 novel cell types (i.e., 100% of the total heldout cell types) (Tab. 16 and Tab. 17). These results show a significant leap from the existing TFMs, which either do not work or only work for annotating a handful of novel types [68, 45, 66].

## 4.4 Marker Gene Prediction

Cell-type-specific genes, or marker genes, are highly expressed in a specific cell type but exhibit low expression in others. These genes play a crucial role in delineating cell functions in diverse tissue contexts. Identifying marker genes in less characterized cell types is an ongoing challenge [52].

**Setup.** We sought to assess whether the pre-trained TFMs can discriminate marker from non-marker genes for any cell type without any supervised fine-tuning. This zero-shot experiment evaluates whether the TFM is able to learn biologically meaningful gene co-expression patterns without supervision. For each cell, we quantified the marker gene potential of each gene by the changes in TFM-generated cell representations after *in-silico* knockout of the target gene (details in App. E.4). Here we assume that the larger the change the higher the marker gene potential. We discussed the

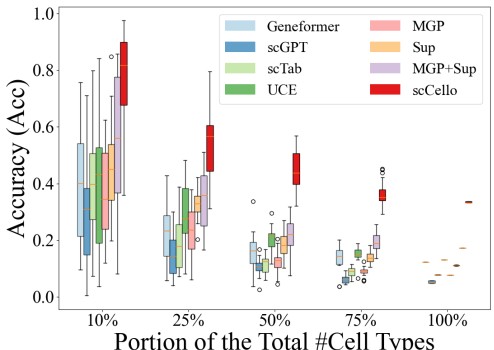

Figure 2: Novel cell type classification on OOD cell type dataset $D_1^{ct}$ for increasing difficulties.

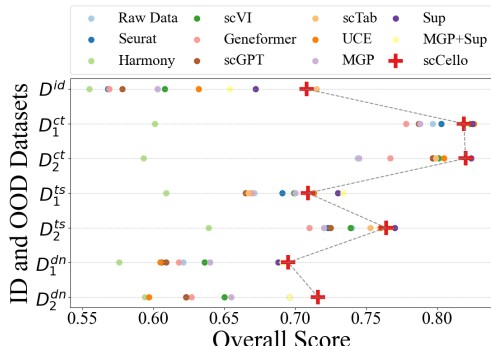

Figure 3: Batch integration on the curated ID and OOD datasets.

caveat of this approach in Sec. 5. As test data, we used GSE96583 [32] ($D_1^{mk}$) and GSE130148 [65] ($D_2^{mk}$) . We obtained the marker gene labels from CellMarker2 [29] and PanglaoDB [21]. We also compared with a non-TFM method, Differential Expression Tests (DET) [60].

**Zero-shot transferability.** In Tab. 3, scCello outperforms other TFMs, improving upon the second-best method by 1.8% in average AUROC. The inclusion of cell label information during pre-training boosts TFM performance, as evidenced by the strong results of scTab, Sup, MGP+Sup and scCello. This is due to the biological correlation between marker genes and cell types. Furthermore, employing cell ontology graphs further improves the prediction accuracy over MGP+Sup.

## 4.5  Cancer Drug Response Prediction

Developing effective drugs for cancer treatment is challenging due to individual variability in drug responses. Accurately predicting cancer drug responses (CDR) can greatly aid anti-cancer drug development and improve our understanding of cancer biology [43].

**Setup.** Following the approach of scFoundation [26], cell representations were extracted from fixed TFMs and integrated into the DeepCDR [43] pipeline to estimate the half-maximal inhibitory concentration ($IC_{50}$) values of drugs (details in App. E.6). We benchmarked our method against DeepCDR, scFoundation, and other TFM baselines, using the same pre-processed data as DeepCDR.

**Zero-shot transferability.** In Tab. 4, scCello is among the top 3 along with scGPT and UCE, achieving 7.4% improvement in PCC over the base method DeepCDR. This highlights scCello's transferability in enhancing specialized task-oriented methods. In particular, it can be used as an powerful feature extractor for diverse downstream tasks.

## 4.6  Batch Integration

The scRNA-seq atlases, assembled from datasets across various labs and conditions, are prone to unwanted technical variations known as batch effects [46]. These effects can significantly affect the generalization ability of TFMs especially because they require pre-training on a massive amount of heterogeneous scRNA-seq data pooled from many studies. Here we sought to evaluate scCello's robustness to batch effects without fine-tuning.

**Setup.** We adopted the same baselines as in zero-shot cell type clustering (Sec. 4.2.1), and followed the evaluation protocol of scGPT [14]. We evaluated on one ID dataset $D^{id}$ and six OOD datasets $D_i^{cond}$ ($cond \in \{ct, ts, dn\}, i \in \{1, 2\}$) (see complete results of all metrics in App. E.7).

**Robustness to data noise.** Fig. 3 shows that scCello excels in 3 out of 7 datasets, and achieves comparable performance on another 3 datasets. The performance is attributable to the use of cell type information as the ablated baseline MGP conferred much lower batch integration score compared to Sup and scCello.

Table 5: Pre-training loss ablation on the cell type clustering and novel cell type classification (*abbr.*, "clf.") tasks.

| Config | Cell Type Clustering | | Novel Cell Type Clf. | |
|---|---|---|---|---|
| | $D_2^{ct}$ AvgBio↑ | $D_2^{dn}$ AvgBio↑ | $D_1^{ct}$ Acc↑ | Macro F1↑ |
| Full Loss | 0.786 | 0.643 | 0.335 | 0.150 |
| *w/o* $\mathcal{L}_{\mathrm{MGP}}$ | 0.774 (↓1.5%) | 0.640 (↓0.5%) | 0.287 (↓14.3%) | 0.131 (↓12.7%) |
| *w/o* $\mathcal{L}_{\mathrm{Inter}}$ | 0.778 (↓1.0%) | 0.620 (↓**3.6%**) | 0.147 (↓**56.1%**) | 0.052 (↓**65.3%**) |
| *w/o* $\mathcal{L}_{\mathrm{Intra}}$ | 0.730 (↓**7.1%**) | 0.626 (↓2.6%) | 0.280 (↓16.4%) | 0.118 (↓21.3%) |
| *w/o* $\mathcal{L}_{\mathrm{Reg}}$ | 0.764 (↓2.8%) | 0.638 (↓0.8%) | 0.296 (↓11.6%) | 0.134 (↓10.7%) |

Table 6: Overall performance *v.s.* the number of parameters.

| Method | Perf. Rank | #Params (M) |
|---|---|---|
| Geneformer | 6.3 | 10.3 |
| scGPT | 6.2 | 51.3 |
| scTab | 4.2 | **9.7** |
| UCE | 4.8 | 674.7 |
| MGP | 6.7 | 10.3 |
| Sup | 3.3 | 10.4 |
| MGP+Sup | 3.2 | 10.9 |
| **scCello** | **1.3** | 10.7 |

## 4.7 Ablation Study

**Ablation of pre-training losses.** Tab. 5 reports the cell type clustering (Sec. 4.2.1) and novel cell type classification (Sec. 4.3) performance of scCello by using full or partial pre-training losses. Removing any of the four losses in Eqn. 5 resulted in decreased performance, corroborating the benefits of the proposed pre-training losses. Notably, removing the inter-cellular ontology relation loss $\mathcal{L}_{\mathrm{Inter}}$ led to 56.1% and 65.3% decrease in terms of Acc. and Macro F1 on novel cell type classification task, respectively. This shows the upmost importance of the structurally induced loss and ultimately the use of cell ontology graph information.

**Parameter efficiency.** Tab. 6 demonstrates that scCello is highly parameter-efficient, utilizing up to 60 times fewer parameters than the largest existing TFM, UCE, while still achieving the best average performance rankings across all downstream tasks. With an average performance rank of 1.3, scCello consistently ranks first or near the top in nearly every task.

**Visualization.** Visualization and analysis of scCello's learned cell representations were presented in App. E.8. In short, biologically similar cell types are closer to each other and farther from those dissimilar ones in the t-SNE 2D space (Fig. 11).

## 5 Discussion and Conclusion

**Limitation and future work.** The cell ontology is constantly revised and expanded. In the future, we plan to investigate more efficient methods for fine-tuning scCello to enable continual learning of updated ontology, rather than retraining the entire model. Additionally, we aim to scale up the model size of scCello to increase its expressiveness and capacity. For the zero-shot marker gene prediction experiments (Sec. 4.4), one caveat is that our in-silico gene knockout approach also detects essential genes such as housekeeping genes [17] and transcription factors that are master regulators [9], which may not necessarily be marker genes. Nonetheless, deletion of these influential genes will also lead to large change of the transcriptome landscape of the cell. We will explore this in future study.

**Societal impact.** This work proposes a novel cell ontology-guided TFM, scCello, to enhance cell representation learning. On the positive side, once pre-trained, scCello can serve as a foundational model capable of facilitating scientific discoveries across various downstream tasks related to cells and cellular processes. However, on the negative side, the pre-training of scCello requires significant computational resources, potentially resulting in substantial carbon dioxide emissions that could contribute to environmental harm.

**Conclusion.** The proposed scCello incorporates cell ontology knowledge into its pre-training process by simultaneously modeling at the gene level, intra-cellular level, and inter-cellular level. We constructed a large-scale cell type identification benchmark to evaluate the model's generalization capabilities, both in-distribution and out-of-distribution. Our evaluation demonstrates that scCello also exhibits strong transferability, as evidenced by its performance on other biologically meaningful downstream tasks such as zero-shot novel cell type classification and cell-type-specific marker gene prediction. Foundational models are typically heavy on the parameters for them to have sufficient capacity to learn from unlabeled data from scratch. This limits their usage to only fine-tuning tasks as pre-training them is prohibitive without large compute. Our proposed approach provides an efficient way of leveraging the prior knowledge at the pre-training, which led to much smaller parameter size while achieving performance comparable of the TFMs that are 5-60 times bigger. Together, scCello is a knowledge-informed and general purpose deep learning model that can be fine-tuned

for a wide array of downstream applications, aiding in the rapid identification of novel cell types, disease-associated genes, and effective cancer drugs.

## Acknowledgments and Disclosure of Funding

The authors would like to thank Chence Shi, Meng Qu, Zhaocheng Zhu, and Sophie Xhonneux for their helpful discussions and comments. We also appreciate all anonymous reviewers for their constructive suggestions. This project is supported by Intel-MILA partnership program, the Natural Sciences and Engineering Research Council (NSERC) Discovery Grant, the Canada CIFAR AI Chair Program, collaboration grants between Microsoft Research and Mila, Samsung Electronics Co., Ltd., Amazon Faculty Research Award, Tencent AI Lab Rhino-Bird Gift Fund and a NRC Collaborative R&D Project (AI4D-CORE-06). This project was also partially funded by IVADO Fundamental Research Project grant PRF-2019-3583139727. Y.L. is supported by Canada Research Chair (Tier 2) in Machine Learning for Genomics and Healthcare (CRC-2021-00547) and Natural Sciences and Engineering Research Council(NSERC) Discovery Grant (RGPIN-2016-05174). The computation resource of this project is supported by Mila, Calcul Québec and the Digital Research Alliance of Canada.

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

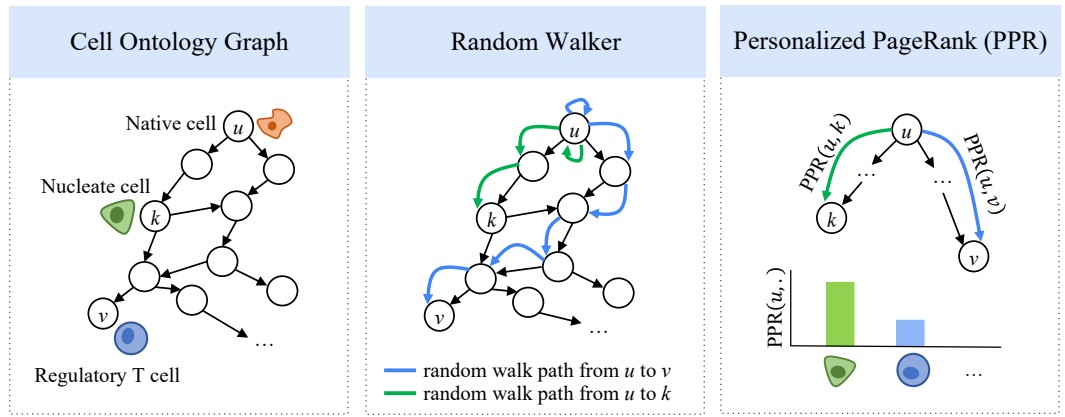

Figure 4: Graphical illustration of applying the Personalized PageRank (PPR) algorithm to cell ontology graph. As explained in App. A, PPR conducts random walks over the ontology graph with respect to a target cell type $u$, and converges to a steady state when the likelihood of terminating on each node stabilizes into a steady distribution. This likelihood distribution determines the final PPR score $\text{PPR}(\cdot)$ and reflects the structural similarity between cell types.

## A PPR Transformation

**Personalized PageRank (PPR).** Personalized PageRank (PPR) extends the classic PageRank algorithm, which Google originally developed to rank web pages in search engines. PageRank conducts this by analyzing large-scale hyperlinked graphs on the web using random walker simulations. Unlike traditional PageRank that assigns a universal score to each web page, PPR customizes these scores. Specifically, individual user preferences during searches are incorporated, so that PPR can focus on web pages particularly relevant to each user. Due to its flexibility and effectiveness, PPR has been widely applied in graph learning across various fields, such as social networks, recommendation systems, and biological data analysis.

As illustrated in Fig. 4, this algorithm starts with a predefined preference node (or target node), which is emphasized according to the user's interests. Subsequently, a random walk is conducted on the graph to facilitate graph traversal. At each step of the walk, there is a fixed probability $\alpha$ that the walker will jump back to the target node from the current node instead of moving to an adjacent node chosen at random. This process of jumping, commonly referred to as "teleportation", biases the walk towards subgraphs that are of particular importance to the target node, thus personalizing the results according to user preferences. The walk continues until it reaches a steady state, at which point the likelihood of being on each node stabilizes into a steady-state distribution. These stabilized

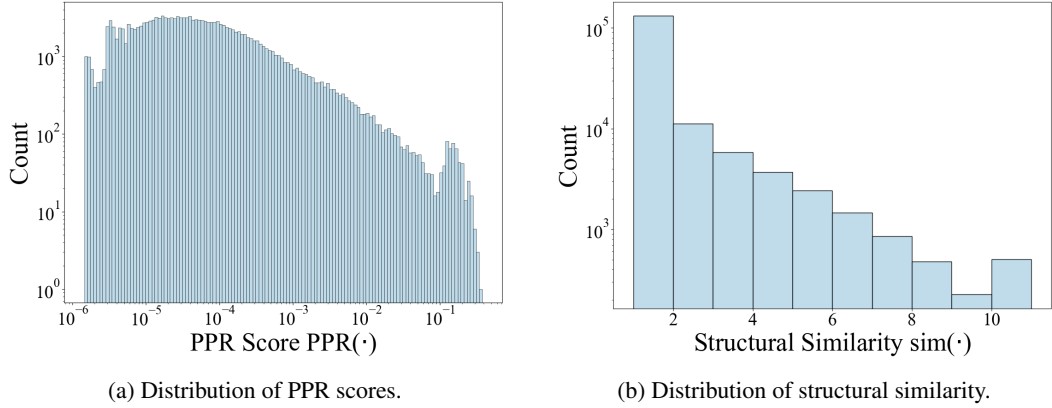

(a) Distribution of PPR scores.

(b) Distribution of structural similarity.

Figure 5: Comparison of the distributions for the PPR scores $\text{PPR}(\cdot)$ and the structural similarity $\text{sim}(\cdot)$ after the transformation.

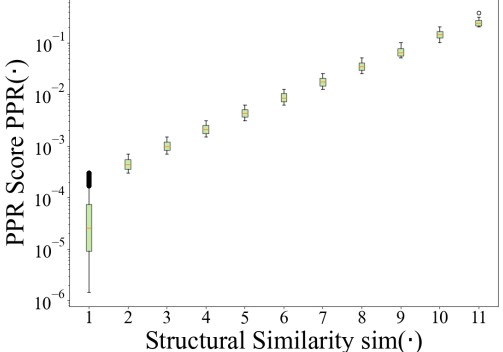
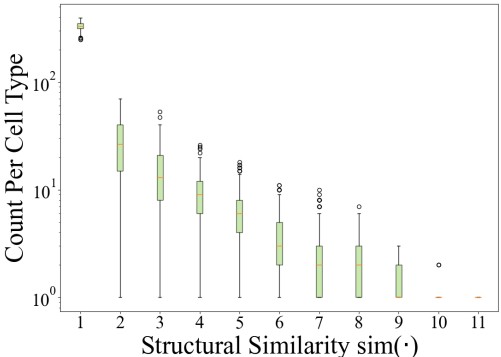

Figure 6: Relationships between the structural similariity sim(·) after PPR transformation and the original PPR scores PPR.

Figure 7: Frequency for each target cell type to be associated with other cell types that is at specific levels of structural similarity.

probabilities, reflecting both the graph's structure and the user's preferences, determine the PPR scores. These scores effectively evaluate each node's structural similarities and rank them according to their relevance and importance from a personalized perspective.

**PPR transformation.** In scCello, the PPR algorithm is applied to the cell ontology graph to assess the structural similarities among cell types, or to measure their importance relative to a specified target cell type. We implemented PPR using the "pagerank" function in NetworkX [23] with "personalization" as arguments.

However, modification is needed to integrate PPR into TFM pre-training. The PPR scores are in real-number format and susceptible to numerical noise. Also, as shown in Fig. 5a, these scores typically exhibit a skewed distribution, concentrated around lower magnitudes. Consequently, setting precise thresholds to differentiate between node similarity and dissimilarity is challenging. Moreover, the vast amount small PPR values may be indistinguishable from noise.

To mitigate the effects of numerical noise and skewed magnitudes for the PPR scores, we employ truncation, logarithmic scaling, and discretization as outlined in Eqn. 3. Note that Eqn. 3 defines a monotonic, non-decreasing function that preserves the relative order between nodes. Its minimum value is set to 1 for the least similar cell types.

This equation transforms the raw PPR score, $\text{PPR}(\cdot)$, into the final structural similarity, $\text{sim}(\cdot)$. This transformation ensures that $\text{sim}(\cdot)$ accurately reflects pronounced similarities as defined by the cell ontology and avoids emphasizing minor dissimilarities that could mislead during TFM pre-training.

**Analyses.** In Fig. 5, we present a comparison of the distributions for the PPR score, $\text{PPR}(\cdot)$, and the transformed structural similarity, $\text{sim}(\cdot)$. After transformation, the distribution of $\text{sim}(\cdot)$ is less skewed and exhibits clear discretization. This facilitates the setting of definitive thresholds for distinguishing between similarity and dissimilarity among cell types, thereby enabling the effective incorporation of the cell ontology graph in scCello's pre-training.

In addition, we provide detailed insights into the scale of structural similarity, the distribution of these similarities for each cell type, and examples of cell types associated with various levels of structural similarity:

(1) Fig. 6 illustrates the correspondence between the structural similarity after PPR transformation and the original PPR scores, showcasing a log-linear relationship as expected. This helps clarify the scaling of structural similarity, which is discretized into integer levels ranging from 1 to 11.

(2) Fig. 7 demonstrates how frequently each target cell type is associated with other cell types at specific levels of structural similarity. Consequently, during scCello's pre-training, a substantial number of negative samples are expected to be utilized in the inter-cellular relational alignment objective, as outlined in Sec. 2.4.

Table 7: Examples of cell types associated with various levels of structural similarity, $\text{sim}(\cdot)$, for specified target cell types. Cell types demonstrated in the cell ontology graph in Fig. 1 are underlined.

| Target Type | $\text{sim}(\cdot)$ | Corresponding Cell Types |
|---|---|---|
| T Cell | 8 | "gamma-delta T cell", "mature T cell", "lymphocyte" |
| | 7 | "mature gamma-delta T cell", "$\alpha$-$\beta$ T cell", "mature $\alpha$-$\beta$ T cell", "thymocyte" |
| | 6 | "B cell", "double-positive, $\alpha$-$\beta$ thymocyte", "CD8-positive, $\alpha$-$\beta$ thymocyte", "CD4-positive, $\alpha$-$\beta$ T cell", "CD8-positive, $\alpha$-$\beta$ T cell", "double negative thymocyte" |
| | 5 | "dendritic cell", "innate lymphoid cell", "plasmablast", "mononuclear cell", "regulatory T cell", "memory T cell", "myeloid leukocyte", "naive T cell", "mature B cell", "CD4-positive, CD25-positive, $\alpha$-$\beta$ regulatory T cell" |
| | . . . | . . . |
| | 1 | "renal intercalated cell", "smooth muscle fiber of ileum", "type II pneumocyte", "hematopoietic cell", "neuron", "common lymphoid progenitor", . . . |
| Neuron | 7 | "secretory cell" |
| | 6 | "glutamatergic neuron", "GABAergic neuron", "motor neuron", "neural cell", "peripheral nervous system neuron" |
| | 5 | "glycinergic neuron", "retinal bipolar neuron", "native cell", "enteric neuron", "retina horizontal cell", "amacrine cell", "neuronal receptor cell" |
| | 4 | "retinal ganglion cell", "endocrine cell", "neuroendocrine cell", "cerebral cortex GABAergic interneuron", "muscle cell", "somatic cell" |
| | . . . | . . . |
| | 1 | "germ cell", "T cell", "tracheal goblet cell", "DN3 thymocyte", "promonocyte", "cerebral cortex endothelial cell", . . . |

(3) Tab. 7 displays examples of highly similar and dissimilar cell types categorized into various levels of structural similarity, specifically targeting "T cell" and "neuron" types.

## B  Data Preprocessing Details

**Download and Preprocessing.**    We downloaded from CellxGene [1] census version 2023-7-25. We focused on 291 datasets for human scRNA-seq. We preprocessed the dataset by the following steps:

(1) **Remove non-primary cells.** Some data on CellxGene was duplicated due to multiple submissions of the same dataset from different research groups, therefore cells marked as "non-primary" were filtered out to prevent label leakage between pre-training and downstream.

(2) **Filter out cells not produced by 10x-based [69] sequencing protocols.** There are numerous sequencing protocols in CellxGene database besides 10x-based sequencing [69], such as Drop-seq [58] and MARS-seq [35]. Only sequencing data from 10x-based sequencing protocols was kept to avoid large variation of data signals [46].

(3) **Exclude cancer cells.** Cancer cells were highly dissimilar to normal cells and even occupied a large amount in the CellxGene database (nearly 12%). These cells could bring unexpected signals and skew the data, therefore we excluded these cancer cells.

To build downstream datasets for out-of-distribution (OOD) generalization evaluation, we first held out two category sets for each of the three settings: unseen cell types, unseen tissues and unseen

Table 8: Data statistics for our curated pre-training and downstream datasets, where the downstream datasets encompass one ID dataset and six OOD datasets under three different OOD scenarios, including unseen cell types, unseen tissues and unseen donors (Sec. 4.1). The blue colored numbers represent disjoint categories of that column. For example, in the "cell type" column, the cell type set in the pre-training data, and the cell type set in the OOD cell type dataset $D_1^{ct}$ and $D_1^{ct}$ are disjoint.

| Dataset | #Total Cells | #Cell Types | #Tissues | #Donors | #Conditions | #Batches |
|---|---|---|---|---|---|---|
| Pre-training data | 22,293,755 | 398 | 140 | 4,103 | 55 | 267 |
| ID dataset $D^{id}$ | 22,317 | 318 | 132 | 3,447 | 54 | 261 |
| OOD cell type dataset $D_1^{ct}$ | 486,810 | 87 | 125 | 122 | 35 | 90 |
| OOD cell type dataset $D_2^{ct}$ | 435,791 | 87 | 128 | 106 | 40 | 117 |
| OOD tissue dataset $D_1^{ts}$ | 335,675 | 186 | 32 | 1,801 | 10 | 28 |
| OOD tissue dataset $D_2^{ts}$ | 341,681 | 205 | 32 | 2,052 | 7 | 25 |
| OOD donor dataset $D_1^{dn}$ | 2,528,134 | 439 | 91 | 525 | 36 | 127 |
| OOD donor dataset $D_2^{dn}$ | 2,521,868 | 404 | 101 | 525 | 33 | 123 |
| In Total | / | 572 | 204 | 5153 | / | / |

donors. Each category set were randomly selected with selection ratios 15%, 15% and 10% for the three OOD settings respectively. During the selection, we prohibited any category associated with more than 0.1% of the total pre-processed cells from being selected. This avoids losing too much data for pre-training. After the selection, cells associated with each held category set are collected, resulting in two OOD downstream datasets for each of the three OOD settings. These datasets are denoted as $\{D_i^{ct}\}_{i=1}^2$ for the OOD **c**ell **t**ype setting, $\{D_i^{ts}\}_{i=1}^2$ for the OOD **tis**sue setting, and $\{D_i^{dn}\}_{i=1}^2$ for the OOD **do**nor setting.

By excluding cells with at least one property belong to any of the six held category sets, the remaining data is further split into 99.9% as our pre-training data and 0.1% as the in-distribution (ID) downstream dataset $D^{id}$. This way, our pre-training data and the ID dataset $D^{id}$ share similar data distributions.

**Data Statistics.** We summarize the data statistics for our curated pre-training dataset, one ID dataset and six OOD datasets in Tab. 8.

## C  Discussion on Ontology Graph Modeling

**Graph Neural Networks (GNNs).** One essential component of scCello is to model the cell ontology prior graph. GNNs are essential for managing graph-structured data by using a process called message passing [36]. This process lets nodes gather information from their neighbors, capturing their local connections and features, as seen in technologies like GCN [39]. Further advancements like GraphSAGE [24] and GAT [64] allow GNNs to handle larger areas of the graph and more complex relationships, useful in fields from social networks to drug discovery.

Recently, combining GNNs with transformer models, like GraphFormer [74], has proven effective. This integration allows the models to process both textual and graph data simultaneously, enhancing the understanding of the graph's structure without needing external measures.

**Why we left GNNs for future work.** While using GNNs in TFM modeling can internalize ontology graph knowledge in TFM modeling and no custom metric like that for PPR transformation is necessary, the cellular ontology graph presents specific challenges that hinder effective direct modeling by GNNs. The key issues is that the ontology graph is extremely sparse with about 2.7k nodes and 3.9k edges, and faces long-distance issues. For example, the average pairwise distance of 398 nodes (i.e., cell types) associated with our pre-training scRNA-seq data is 7.39, and the maximum distance is 18. Therefore, it requires multiple layers of graph propagations (around 7 layers), risking over-smoothing [57] of cell type representations.

Table 9: Hyper-parameters comparison between TFM baselines (introduced in Sec. 4.1) and our TFM scCello. "The number of" is denoted with the symbol #.

| Configuration | Geneformer [61] | scGPT [14] | scTab [19] | UCE [55] | scCello |
|---|---|---|---|---|---|
| #Parameters | 10,316,196 | 51,330,049 | 9,655,628 | 674,745,857 | 10,683,654 |
| Total GPUs | 12 * V100 (32G) | 4 * A100 | 1 * A100 | 24 * A100 (80G) | 4 * A100 (40G) |
| Training Time | 3 days | 3 days | / | 43.5 days | 2 days |
| Sequence Length | 2,048 | 1,200 | 19,331 | 1,024 [N] | 2,048 |
| Gene Mask Ratio | 15% | / | / | 20% | 15% |
| Batch Size Per GPU | 12 | 32 | 2,048 | 6 | 12 |
| Gradient Accumulation Steps | 1 | 1 | 1 | 4 | 4 |
| Effective Batch Size | 144 | 128 | 2048 | 576 | 192 |
| Cell Reprs. | Avg. pooling | CLS | / | CLS | CLS |
| #Genes in Token Vocabulary | 25,424 | 48,292 | 19,331 | Any protein-coding genes | 25,424 |
| #Transformer Layers | 6 | 12 | / | 33 | 6 |
| Transformer Layer Hidden Dimension | 512 | 512 | / | 5,120 | 512 |
| Transformer Layer Embedding Size | 256 | 512 | / | 1,280 | 256 |
| #Transformer Heads | 4 | 8 | / | 20 | 4 |
| Transformer Layer Activation Function | GeLU | ReLU | / | ReLU | ReLU |
| MLP Layer Activation Function | ReLU | ReLU | / | GeLU | ReLU |
| Dropout | 0.02 | 0.2 | / | 0.05 | 0.02 |

# D   Implementation Details

**scRNA-seq Data.**   scRNA-seq can enable the quantification of gene expression profiles of individual cells. Each cell's gene expression profile can be described by the set $\hat{X} = \{(e_1, g_1), (e_2, g_2), \ldots, (e_M, g_M)\}$, where $e_k$ denotes the expression count of gene $g_k$, with $e_k \geq 0$. A value of $e_k = 0$ indicates that the gene $g_k$ is not expressed or not detected by the sequencing experiment. We use the same gene vocabulary set as [61], with the number of genes $M = 25,424$.

**Gene Token Vocabulary.**   The gene vocabulary set contains both protein-coding genes and miRNA genes. $M$, the number of genes, is not the same as the number of all tokens in the model vocabulary. scCello has $M$ gene tokens plus three more special tokens [MASK] for masking, [CLS] for the start of a sentence and [PAD] for padding.

**Rank Value Encoding.**   Unlike natural languages, which inherently follow a sequential order, scRNA-seq data presents a unique challenge due to the lack of intrinsic order among gene tokens. Therefore, we employ Rank Value Encoding [61] approach to rank genes based on their normalized expression set $\{(\tilde{e}_i, g_i)\}_{i=1}^{M}$. Specifically, gene expressions are first normalized by the total count within a cell [71] in a cell-wise manner, and then normalized through gene-specific weighting factors in a gene-wise manner. These factors are adopted from [61], which calculates the non-zero median value of expression of each detected gene across all cells. By design, these factors are assigned to emphasize lowly-expressed but essential genes, such as transcription factors [41], while deprioritizing ubiquitously expressed housekeeping genes [17].

After the normalization and ranking, it results in an ordered sequence of gene identities $X = [g_{\pi(1)}, g_{\pi(2)}, \ldots, g_{\pi(M)}]$ with an index permutation $\pi(\cdot)$, satisfying $\tilde{e}_{\pi(1)} \geq \tilde{e}_{\pi(2)} \geq \cdots \geq \tilde{e}_{\pi(M)}$. To mitigate memory consumption, zero-expressed genes are removed and the gene sequence is

Table 10: Metrics used in downstream tasks.

| Task | Metrics |
|------|---------|
| Cell Type Clustering (Sec. 4.2.1) | NMI, ARI, ASW, AvgBio |
| Cell Type Classification (Sec. 4.2.2) | Acc, Macro F1, AvgBio, $\Delta_{\text{AvgBio}}$ |
| Novel Cell Type Classification (Sec. 4.3) | Acc, Macro F1 |
| Marker Gene Prediction (Sec. 4.4) | AUROC |
| Cancer Drug Response Prediction (Sec. 4.5) | PCC |
| Batch Integration (Sec. 4.6) | NMI, ARI, ASW, AvgBio, $\text{ASW}_b$, GraphConn, AvgBatch, Overall |

further truncated with a context length $L=2,048$ in practice. This rank-based approach offers better robustness against technical artifacts than directly using the original numerical expressions, which can vary significantly in magnitude across different experimental assays [46].

**Cell and Cell Type Representations.** Given a pre-training dataset with $N$ cells $\mathcal{X} = \{X_1, X_2, \ldots, X_N\}$, each cell $X_i$ can be mapped to a specific cell type ontology identifier $c_i \in \mathcal{V}$. For analyzing, scCello denotes cell $X_i$'s representation as $\boldsymbol{z}_i$ and cell type $c_i$'s representation as $\boldsymbol{h}_{c_i}$.

**Masked Gene Prediction.** Given a batch of cells $\{X_i\}_{i=1}^B$, scCello predicts a gene token $g_k$ based on the ordered gene sequence context $X_{i,\setminus k}=[g_1, \ldots, g_{k-1}, [\text{MASK}], g_{k+1} \ldots, g_M]$ after replacing the token with a special [MASK]. This objective (term as $\mathcal{L}_{\text{MGP}}$) aims to capture complex but important gene-gene interactions within one cell, like regulatory mechanisms between transcription factors and other genes:

$$\mathcal{L}_{\text{MGP}} = -\sum_{k=1}^B \mathbb{E}_{i \sim \Psi} - \log p(x_i | X_{k,\setminus i}) \tag{6}$$

where tokens are masked by a pre-defined distribution $\Psi$, same as that in BERT [15]. Specifically, 80% selected genes are replaced with [MASK], 10% selected genes are kept the same as its original, and 10% selected genes are replaced with random gene tokens.

**Model Architecture.** scCello utilizes a stack of self-attention transformer encoder layers [63], eacg composed of a self-attention and feedforward neural networks. The self-attention mechanism processes the input sequence, effectively capturing interactions between gene tokens.

**Configuration Hyper-parameters.** Besides scCello, we also summarize essential hyper-parameters for TFM baselines in Tab. 9 for comparison. It includes pre-training configurations like batch size, sequence length, and training time consumed. It also includes architecture configurations for the transformer model backbone, such as the number of transformer layers and the embedding size of transformer layers. Note that scTab uses TabNet [3] instead of transformer layers as model backbone, therefore its architecture configurations are not recorded in the table.

# E  Downstream Experiment Details

## E.1  Evaluation Metrics

All metrics used in downstream tasks are summarized in Tab. 10 and introduced below.

**Normalized Mutual Info Score (NMI).** The NMI is a metric that quantifies the similarity between two differen clustering assignments or labelings of the same set of samples. We use NMI to compare the cell-type labels, with the cluster indices obtained from applying the Louvain clustering algorithm [12] on the target dataset.

We denote the two label assignments of the same $N$ cell samples as $C$ and $K$, representig the cell-type labels and the Louvain cluster indices, respectively. The entropy of a label assignment, say $C$, is a

measure of the uncertainty associated with that assignment set. It's calculated as:

$$H(C) = -\sum_{i=1}^{|C|} P(i) \log P(i) \tag{7}$$

where $|C|$ is the number of unique cell types and $P(i) = \frac{|C_i|}{N}$ is the probability that a randomly selected sample belongs to the class $C_i$. The entropy $H(K)$ for the cluster indices $K$ is computed similarly, with $Q(j) = \frac{|K_j|}{N}$ being the probability of a sample belonging to the cluster $K_j$:

$$H(K) = -\sum_{j=1}^{|K|} Q(j) \log Q(j) \tag{8}$$

The mutual information (MI) between $C$ and $K$ qunatifies the amount of information shared between the two label assignments. It is calculated by:

$$\text{MI}(C, K) = \sum_{i}^{|C|} \sum_{j}^{|K|} R(i,j) \log \frac{R(i,j)}{P(i)Q(j)} \tag{9}$$

where $R(i,j) = \frac{|C_i \cap K_j|}{N}$ is the probability that a randomly selected sample belongs to both the class $C_i$ and the cluster $K_j$.

The normalized mutual information (NMI) is defined as:

$$\text{NMI}(C, K) = \frac{\text{MI}(C, K)}{\text{mean}(H(C), H(K))} \tag{10}$$

NMI is a normalized version of MI, scaled by the mean of the entropy terms for cell-type labels and cluster indices. This normalization ensures that NMI values range from 0 to 1, where 0 indicates no correlation between the two label assignments, and 1 represents a perfect match.

To obtain the best match between the clusters and the cell-type labels, we performed optimized Louvain clustering over a range of resolutions from 0.1 to 2, in steps of 0.1. The clustering output with the highest NMI score, when compared to the cell-type label set, was selected as the optimal clustering result. The implementation of NMI used in this study was from the scib python library [47].

**Adjusted Rand Index Score (ARI).**   The ARI is another metric used to evaluate the similarity between the clustering assignment and the cell type labels of the same set of samples, similar to the NMI metric. In this context, we similarly denote the cell-type labels as $C$ and the Louvain [12] cluster indices computed on the target dataset as $K$.

The Rand Index (RI) is a measure of the overlap between the two clusterings, $C$ and $K$. It considers both the correct clustering overlaps and the correct disagreements between the two clusterings [54]. Formally, if we define $a$ as the number of pairs of elements that belong to the same set in both $C$ and $K$, and $b$ as the number of pairs of elements that are in different sets in $C$ and in different sets in $K$, the unadjusted RI is given by:

$$\text{RI} = \frac{a + b}{C_2^N} \tag{11}$$

where $N$ is the total number of cell samples and $C_2^N$ represents the total number of possible pairs in the dataset.

However, the unadjusted RI does not account for the possibility of random label assignments leading to correct overlaps by chance. To address this issue, the adjusted RI (ARI) is introduced, which corrects for randomly correct labels by discounting the expected RI of random labelings:

$$\text{ARI} = \frac{\text{RI} - \mathbb{E}[\text{RI}]}{\max(\text{RI}) - \mathbb{E}[\text{RI}]} \tag{12}$$

The ARI ranges from 0 to 1, where 0 corresponds to a random labeling, and 1 indicates a perfect match between the two clustering assignments.

Similar to NMI, we performed NMI-optimized Louvain clustering to obtain the best match between the clusters and the cell-type labels. Specifically, we executed Louvain clustering over a range of resolutions and selected the clustering output with the highest NMI score when compared to the cell type label set. The implementation of ARI used in this study was from the scib python library [47].

**Average Silhouette Width Score (ASW).** The silhouette width [56] is a metric that evaluates the quality of a clustering solution by quantifying the relationship between the within-clustering distances and the between-cluster distances for each data point. Like the NMI and the ARI, the silouette calculates the similarity between the clustering assignment and the cell type labels of the same set of samples.

For each cell sample, the silhouette width is computed based on two scores: (1) $a$: the mean distance between a sample and all other samples in the same cluster; and (2) $b$ the mean distance between a sample and all samples in the nearest neighboring cluster. The silhouette score $s_i$ for each sample $i$ is defined as

$$s_i = \frac{b - a}{\max(a, b)} \tag{13}$$

The silhouette score ranges from -1 to 1, with higher values indicating that the sample is well-matched to its own cluster and dissimilar to the nearest neighboring cluster.

To obtain an overall assessment of the clustering quality, the average silhouette width (ASW) is calculated by averaging the silhouette scores $s_i$ across all samples. This overall ASW, denoted as $\text{ASW}_o$, ranges between -1 and 1, with the following interpretations:

- $\text{ASW}_o$ close to 1: The clusters are dense and well-separated.
- $\text{ASW}_o$ around 0: The clusters overlap, and the between-cluster and within-cluster variability are approximately equal.
- $\text{ASW}_o$ near -1: Strong misclassification has occurred, where the within-cluster variability is greater than the between-cluster variability.

To ensure that the final ASW metric falls within the range of 0 to 1, a scaling operation is often applied:

$$\text{ASW} = \frac{\text{ASW}_o + 1}{2} \tag{14}$$

This scaled ASW value, ranging from 0 to 1, provides a convenient measure for evaluating the quality of the clustering solution, with higher values indicating better separation and cohesion of the clusters.

**AvgBio.** This score combines the three clustering metrics: NMI, ARI and ASW.

$$\text{AvgBio} = \frac{1}{3}(\text{NMI} + \text{ARI} + \text{ASW}) \tag{15}$$

**Silhouette Variant Score ($\text{ASW}_b$).** To evaluate the effectiveness of the batch integration task (Sec. 4.6), a variant of the average silhouette width score (ASW) is employed, referred to as the $\text{ASW}_b$. Unlike $ASW$ based on cell type labels, $\text{ASW}_b$ considers batch labels. This score is designed to assess the degree of batch mixing, where a score of 0 indicates well-mixed batches, and deviations from 0 suggest the presence of a batch effect.

We take the absolute value of the original silhouette width score $\tilde{s}_i$ for sample $i$ based on batch labels:

$$s_i' = |\tilde{s}_i| \tag{16}$$

To ensure higher scores indicate better batch mixing, these scores are scaled by subtracting them from 1. As we expect batches to integrate within cell identity clusters, we compute the $\text{ASW}_{b,j}$ score for each cell label $j$ separately, using the following equation:

$$\text{ASW}_{b,j} = \frac{1}{|C_j|} \sum_{i \in C_j} 1 - s(i)' \tag{17}$$

where $C_j = \{i|c_i = j\}_{i=1}^N$ is the set of cell indices whose cell type label is exactly $j$.

To obtain the final ASW$_b$ score, the label-specific ASW$_{b,j}$ scores are averaged across the set of unique cell type labels:

$$\text{ASW}_b = \frac{1}{|\mathcal{V}|} \sum_{j \in \mathcal{V}} \text{ASW}_{b,j} \tag{18}$$

where $\mathcal{V}$ represents the set of unique cell type labels.

**Graph Connectivity (GraphConn).** The GraphConn metric is designed to assess whether the $k$-nearest neighbor ($k$NN) graph representation of the integrated data directly connects all cells with the same cell type label. This metric operates on the $k$NN graph, denoted as $G_{k\text{NN}}$, which is pre-processed by the Scanpy library using the "scanpy.pp.neighbors" function.

For each cell type label $v \in \mathcal{V}$, where $\mathcal{V}$ represents the set of cell type labels (Sec. 2), a subset $k$NN graph $G_{k\text{NN}}(\mathcal{V}_v; \mathcal{E}_v)$ is created. This subset graph contains only cells from the given label $v$.

Using these subset kNN graphs, the GraphConn score is computed as follows:

$$\text{GraphConn} = \frac{1}{|\mathcal{V}|} \sum_{v \in \mathcal{V}} \frac{|\text{LCC}(G_{k\text{NN}}(\mathcal{V}_v, \mathcal{E}_v))|}{|\mathcal{V}_v|} \tag{19}$$

Here, $|\text{LCC}(\cdot)|$ is the number of nodes in the largest connected component of the graph and $|\mathcal{V}_v|$ is the number of nodes with cell type $v$.

The resultant GraphConn score has a range of $(0; 1]$, where a score of 1 indicates that all cells with the same cell type are connected in the integrated $k$NN graph. The lowest possible score indicates a graph where no cell is connected to any other cell.

It's important to note that the GraphConn score is computed directly on the kNN graph representation of the integrated data. As a result, this metric can be used to evaluate the quality of any integration output, regardless of the specific integration method used.

**AvgBatch.** This score combines two metrics: ASW$_b$ and GraphConn.

$$\text{AvgBatch} = \frac{1}{2}(\text{ASW}_b + \text{GraphConn}) \tag{20}$$

**Overall.** We follow scGPT [14] to calculate a weighted average score of both the batch removal score ASW$_b$ and the bio-conservation score AvgBio to balance biological relevance and batch consistency, following the equation:

$$\text{Overall} = 0.6 * \text{AvgBio} + 0.4 * \text{AvgBatch} \tag{21}$$

**Accuracy (Acc).** In classification tasks like cell type classification (Sec. 4.2.2) and novel cell type classification (Sec. 4.3), we denote the predicted values of the $i$-th sample as $\hat{y}_i$ and the corresponding true label as $y_i$. Then the accuracy metric is defined as

$$\text{Acc}(y, \hat{y}) = \frac{1}{N} \sum_{i=1}^N \mathbb{1}[\hat{y}_i == y_i] \tag{22}$$

where the $\mathbb{1}[\cdot]$ is the indicator function.

**Macro F1 Score (Macro F1).** The F1 Score is essentially defined for binary classification tasks.

$$F_1 = \frac{2}{\text{Recall}^{-1} + \text{Precision}^{-1}} \tag{23}$$

$$\text{Recall} = \frac{\text{TP}}{\text{TP} + \text{FN}} \tag{24}$$

$$\text{Precision} = \frac{\text{TP}}{\text{TP} + \text{FP}} \tag{25}$$

where TP is the number of true positives, FN the number of false negatives, and TP the number of false positives. The recall is intuitively the ability of the classifier to find all the positive samples; The precision is intuitively the ability of the classifier not to label as positive a sample that is negative. For multi-class classification, macro F1 is defined as the average F1 taken over all different classes.

**ROC AUC Score (AUROC).** The Area Under the Receiver Operating Characteristic (AUROC) curve is a metric commonly used to evaluate the performance of binary classification models. It provides a comprehensive measure of the trade-off between the true positive rate (sensitivity) and the false positive rate (1 - specificity) across different classification thresholds.

In a binary classification task, the model's output is typically a probability or score that represents the likelihood of a sample belonging to the positive class. By varying the classification threshold, different operating points on the ROC curve can be obtained, where each point represents a specific combination of true positive rate (TPR) and false positive rate (FPR).

The ROC curve is created by plotting the TPR (y-axis) against the FPR (x-axis) for different classification thresholds. The AUROC is then calculated as the area under this ROC curve, providing a single scalar value that summarizes the overall performance of the binary classifier. The AUROC ranges from 0 to 1, with the following interpretations: (1) AUROC=1 indicates perfect classification, where the classifier can perfectly distinguish between the positive and negative classes; (2) AUROC=0.5 indicates random guessing, indicating that the classifier performs no better than a random prediction.

The AUROC is a widely used metric because it provides a comprehensive evaluation of the classifier's performance across all possible classification thresholds. It is invariant to class imbalance and does not require choosing a specific threshold, making it a robust and threshold-agnostic measure.

Furthermore, the AUROC has a statistical interpretation as the probability that a randomly chosen positive instance will have a higher predicted probability than a randomly chosen negative instance, which provides a clear interpretation of the metric's value.

**Pearson correlation coefficient score (PCC).** The PCCis a widely used measure of the linear relationship between two variables. It quantifies the strength and direction of the linear association between the variables, ranging from -1 to 1. The formula for the PCC between two variables, A and B, is given by:

$$r_{AB} = \frac{\sum_{i=1}^{n}(A_i - \overline{B})(B_i - \overline{B})}{\sqrt{\sum_{i=1}^{n}(A_i - \overline{B})^2}\sqrt{\sum_{i=1}^{n}(B_i - \overline{B})^2}}$$

where $A_i$ and $B_i$ are the individual observations of variables A and B, respectively. $\overline{A}$ and $\overline{B}$ are the sample means of A and B, respectively. $n$ is the number of observations.

The numerator represents the covariance between A and B, which measures how much A and B vary together from their respective means. The denominator normalizes the covariance by the product of the standard deviations of A and B, ensuring that the correlation coefficient falls within the range of -1 to 1. The interpretation of this PPC metric is as follows: (1) $r_{AB}=1$ indicates perfect positive linear correlation (as A increases, B increases proportionally); (2) $r_{AB}= -1$ indicates perfect negative linear correlation (as A increases, B decreases proportionally); (3) $r_{AB}=0$ indicates no linear correlation between A and B; (4) $0 < |r_{AB}| < 1$ indicates that the strength of the linear correlation increases as the value approaches 1 (either positive or negative).

In the context of regression analysis, computing the PCC between each regressor (independent variable) and the target variable can provide insights into the linear relationships between the predictors and the response variable.

### E.2 Cell Type Identification

#### E.2.1 Zero-shot Identification (*i.e.*, Cell Type Clustering)

**Method.** We here discuss the experimental details for Sec. 4.2.1. Cell representations extracted from each baseline model are used to compute the $k$ nearest neighbor ($k$NN) graph using Scanpy's standard protocols [71]. These representations and the $k$NN graph are then processed with Louvain clustering algorithms at various resolutions, ranging from 0.1 to 2 in steps of 0.1. The optimized clustering result is determined by the highest gained NMI score achieved across all the resolutions.

For implementation, we accelerated Louvain clustering by adopting RAPIDS, a software library that enhances data science pipelines by entirely utilizing NVIDIA GPUs instead of traditional CPUs. Additionally, we conducted ten iterations of dataset down-sampling and reported the averaged NMI,

Table 11: Full results for the OOD unseen cell type datasets $D_1^{ct}$ and $D_2^{ct}$ in the ell type clustering.

| Method | OOD CellType Data ($D_1^{ct}$) | | | | OOD CellType Data ($D_2^{ct}$) | | | |
|---|---|---|---|---|---|---|---|---|
| | NMI↑ | ARI↑ | ASW↑ | AvgBio↑ | NMI↑ | ARI↑ | ASW↑ | AvgBio↑ |
| **Non-TFM Methods** | | | | | | | | |
| Raw Data | 0.864 | 0.718 | 0.529 | 0.703 | 0.823 | 0.557 | 0.505 | 0.629 |
| Seurat | 0.893 | 0.773 | 0.590 | 0.752 | 0.884 | 0.723 | 0.605 | 0.737 |
| Harmony | 0.553 | 0.241 | 0.432 | 0.432 | 0.594 | 0.248 | 0.411 | 0.417 |
| scVI | **0.905** | 0.797 | 0.577 | 0.760 | 0.889 | 0.709 | 0.577 | 0.725 |
| **Ontology-Agnostic TFMs** | | | | | | | | |
| Geneformer | 0.846 | 0.697 | 0.525 | 0.689 | 0.846 | 0.629 | 0.530 | 0.668 |
| scGPT | 0.866 | 0.705 | 0.551 | 0.707 | 0.873 | 0.724 | 0.564 | 0.720 |
| scTab | 0.886 | **0.807** | 0.584 | 0.759 | 0.867 | 0.754 | 0.557 | 0.726 |
| UCE | **0.902** | **0.802** | 0.612 | **0.772** | 0.892 | 0.695 | **0.635** | 0.741 |
| MGP | 0.860 | 0.710 | 0.573 | 0.714 | 0.881 | 0.745 | 0.595 | 0.740 |
| Sup | 0.892 | 0.787 | 0.621 | **0.767** | **0.910** | 0.793 | 0.622 | 0.775 |
| MGP+Sup | 0.888 | 0.775 | 0.611 | 0.758 | 0.901 | 0.779 | 0.611 | 0.764 |
| **Ontology-Enhanced TFMs** | | | | | | | | |
| **scCello** | 0.887 | 0.781 | **0.640** | **0.769** | 0.909 | **0.817** | **0.632** | **0.786** |

ARI, ASW, and AvgBio scores. This approach significantly reduced the time required to evaluate a dataset, such as $D^{id}$, from days to just a few minutes.

**Datasets.** As introduced in Sec. 4.2.1, we evaluate one ID dataset ($D^{id}$) and six OOD datasets ($D_i^{cond}$ with $cond \in \{ct, ts, dn\}$ and $i \in \{1, 2\}$) to demonstrate our model's generalization capabilities. These evaluations address various scenarios involving unseen cells for comprehensive testing, including cells with distributions similar to our pre-training dataset, as well as those associated with unseen cell types, tissues, and donors.

**Hyper-parameters.** We used $k = 15$ neighbors to compute the $k$NN graph, with node distances calculated using the euclidean distance between cell representations. The Louvain clustering used seed 0 as the random state and treated the $k$NN graph as unweighted and directed.

**Performance.** In Sec. 4.2.1, Tab. 1 reports only the AvgBio metric for six OOD datasets due to space constraints. Full metrics, including NMI, ARI, and ASW, are detailed in: (1) Tab. 11 for the two OOD cell type datasets ($D_1^{ct}$ and $D_2^{ct}$); (2) Tab. 12 for the two OOD tissue datasets ($D_1^{ts}$ and $D_2^{ts}$); and (3) Tab. 13 for the two OOD donor datasets ($D_1^{dn}$ and $D_2^{dn}$).

### E.2.2 Identification with Fine-tuning (*i.e.*, Cell Type Classification)

**Method.** In this setting, the TFMs are further fine-tuned by adding a simple linear layer atop their model backbones, which transforms the hidden representations into prediction logits. The dimensions of these logits correspond to the number of cell type classes predicted. Importantly, all model parameters, including those of the TFM backbone and the newly added linear layer, are trainable during fine-tuning. The model checkpoint that achieves the highest Macro F1 score on the validation data is then selected for final testing.

**Datasets.** We fine-tuned TFMs on a subset of our curated pre-training data, randomly selecting 90% for training and using the remaining 10% for validation. The final performance was tested on the ID dataset $D^{id}$, which consists of cell samples never seen during scCello 's pre-training. We explored two subset sizes, 0.1% and 1% of the pre-training data, to simulate scenarios where 10× more annotated data becomes available. This exploration is meaningful for real-world applications, where annotating data is both costly and time-consuming.

Table 12: Full results for the OOD unseen tissue datasets $D_1^{ts}$ and $D_2^{ts}$ in the cell type clustering.

| Method | OOD Tissue Data ($D_1^{ts}$) | | | | OOD Tissue Data ($D_2^{ts}$) | | | |
|---|---|---|---|---|---|---|---|---|
| | NMI↑ | ARI↑ | ASW↑ | AvgBio↑ | NMI↑ | ARI↑ | ASW↑ | AvgBio↑ |
| **Non-TFM Methods** | | | | | | | | |
| Raw Data | 0.733 | 0.405 | 0.481 | 0.540 | 0.800 | 0.585 | 0.508 | 0.631 |
| Seurat | 0.777 | 0.497 | 0.488 | 0.587 | 0.813 | 0.560 | 0.535 | 0.636 |
| Harmony | 0.649 | 0.302 | 0.436 | 0.462 | 0.684 | 0.400 | 0.460 | 0.515 |
| scVI | 0.774 | 0.443 | 0.516 | 0.577 | 0.816 | 0.550 | 0.537 | 0.634 |
| **Ontology-Agnostic TFMs** | | | | | | | | |
| Geneformer | 0.736 | 0.412 | 0.468 | 0.539 | 0.787 | 0.499 | 0.505 | 0.597 |
| scGPT | 0.739 | 0.407 | 0.486 | 0.544 | 0.794 | 0.556 | 0.531 | 0.627 |
| scTab | 0.754 | 0.492 | 0.515 | 0.515 | 0.815 | 0.616 | 0.541 | 0.657 |
| UCE | **0.787** | 0.476 | 0.531 | 0.598 | **0.836** | 0.610 | 0.562 | 0.670 |
| MGP | 0.766 | 0.472 | 0.491 | 0.576 | 0.802 | 0.544 | 0.537 | 0.628 |
| Sup | **0.788** | 0.502 | 0.527 | 0.605 | **0.838** | 0.621 | 0.580 | 0.680 |
| MGP+Sup | **0.789** | **0.518** | 0.524 | **0.610** | 0.833 | 0.612 | 0.573 | 0.672 |
| **Ontology-Enhanced TFMs** | | | | | | | | |
| **scCello** | **0.784** | **0.519** | **0.534** | **0.612** | **0.839** | **0.675** | **0.601** | **0.705** |

Table 13: Full results for the OOD unseen donor datasets $D_1^{dn}$ and $D_2^{dn}$ in the Cell Type Clustering. Note that scTab is OOM on these two datasets.

| Method | OOD Donor Data ($D_1^{dn}$) | | | | OOD Donor Data ($D_2^{dn}$) | | | |
|---|---|---|---|---|---|---|---|---|
| | NMI↑ | ARI↑ | ASW↑ | AvgBio↑ | NMI↑ | ARI↑ | ASW↑ | AvgBio↑ |
| **Non-TFM Methods** | | | | | | | | |
| Raw Data | 0.665 | 0.247 | 0.462 | 0.458 | 0.665 | 0.251 | 0.462 | 0.460 |
| Seurat | 0.691 | 0.294 | 0.413 | 0.466 | 0.711 | 0.335 | 0.420 | 0.489 |
| Harmony | 0.679 | 0.286 | 0.405 | 0.456 | 0.690 | 0.324 | 0.408 | 0.474 |
| scVI | 0.699 | 0.269 | 0.466 | 0.478 | 0.722 | 0.311 | 0.471 | 0.502 |
| **Ontology-Agnostic TFMs** | | | | | | | | |
| Geneformer | 0.666 | 0.303 | 0.434 | 0.468 | 0.686 | 0.327 | 0.433 | 0.482 |
| scGPT | 0.656 | 0.259 | 0.452 | 0.456 | 0.677 | 0.298 | 0.456 | 0.477 |
| scTab | / | / | / | OOM | / | / | / | OOM |
| UCE | 0.718 | 0.245 | 0.491 | 0.485 | 0.737 | 0.284 | 0.496 | 0.506 |
| MGP | 0.713 | 0.294 | 0.457 | 0.488 | 0.734 | 0.359 | 0.462 | 0.518 |
| Sup | 0.754 | 0.357 | 0.545 | 0.552 | 0.768 | 0.395 | 0.556 | 0.573 |
| MGP+Sup | 0.754 | 0.373 | 0.532 | 0.553 | 0.768 | 0.398 | 0.544 | 0.570 |
| **Ontology-Enhanced TFMs** | | | | | | | | |
| **scCello** | **0.774** | **0.426** | **0.625** | **0.608** | **0.794** | **0.486** | **0.649** | **0.643** |

**Hyper-parameters.** For scCello, we set the following hyper-parameters for fine-tuning: a learning rate of $5.0 \times 10^{-5}$, a linear learning rate scheduler with 500 warmup steps, a weight decay of 0.001, and a batch size of 24. The same fine-tuning configuration was applied to the three ablation TFMs pre-trained using scCello's codebase (MGP, Sup, and MGP+Sup). For other TFM baselines, we searched for the optimal learning rate to report the final performance.

**Performance.** In Sec. 4.2.2, we reported classification and clustering metrics for TFMs fine-tuned with the 0.1$ subset of the pre-training data. Here, we extend our reporting to TFMs fine-tuned with 1% of a pre-training subset that is $10 \times$ larger. We compare performances at these two subset selection ratios in Tab. 14. We observe that,

Table 14: Cell type identification with fine-tuning evaluated on the ID dataset $D^{id}$, as the pre-training subset data size for fine-tuning increases from 0.1% to 1% for the subset selection ratio.

| Methods | Cell Type Classification | | Cell Type Clustering |
| --- | --- | --- | --- |
| | Acc↑ (0.1% → 1%) | Macro F1↑ (0.1% → 1%) | AvgBio↑ (0.1% → 1%) |
| **Ontology-Agnostic TFMs** | | | |
| Geneformer | $0.747 \to 0.872$ | $0.440 \to 0.664$ | $0.439 \to 0.469$ |
| scGPT | $0.712 \to 0.862$ | $0.344 \to 0.636$ | $0.477 \to 0.481$ |
| scTab | $0.778 \to 0.773$ | $0.373 \to 0.455$ | $0.606 \to 0.589$ |
| MGP | $0.722 \to 0.861$ | $0.287 \to 0.639$ | $0.607 \to 0.631$ |
| Sup | $0.812 \to \underline{0.902}$ | $0.363 \to 0.718$ | $\underline{0.659} \to \underline{0.668}$ |
| MGP+Sup | $\underline{0.820} \to \underline{0.902}$ | $\underline{0.406} \to \underline{0.735}$ | $0.607 \to 0.667$ |
| **Ontology-Enhanced TFMs** | | | |
| **scCello** | $\mathbf{0.867 \to 0.910}$ | $\mathbf{0.511 \to 0.761}$ | $\mathbf{0.694 \to 0.699}$ |

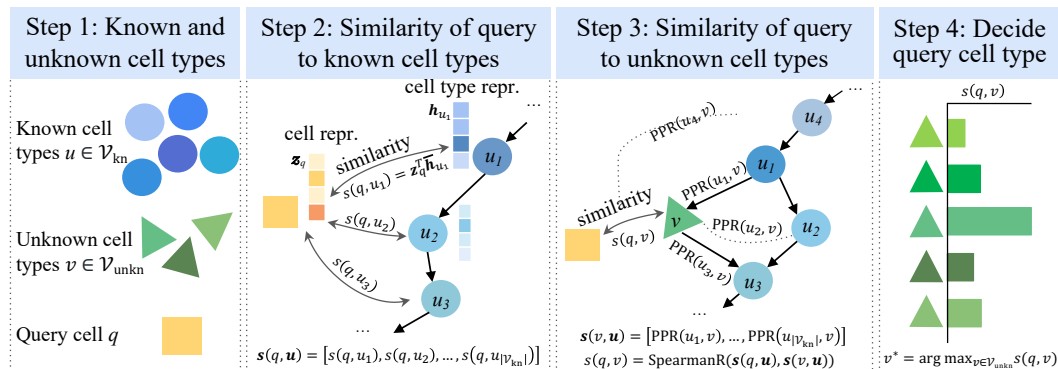

Figure 8: Graphical illustration of our approach for classifying novel cell types (*i.e.*, unknown cell types) (introduced in App. E.3).

(1) As the size of fine-tuning data increases, all TFMs except scTab show benefits and scCello achieves 48.9% improvement in Macro F1 when the data size gets $10\times$ larger. scTab's underperformance may be related to its model capacity, as it employs a TabNet architecture [3]—unlike others that use the powerful standard Transformers [63].

(2) Across both the classification and clustering metrics, scCello's prevails other TFM baselines by a large margin. Remarkably, even when fine-tuned with a smaller 0.1% pre-training subset, scCello surpasses TFMs fine-tuned with a much larger 1% subset, achieving a 3.9% improvement over the best baseline. This underscores scCello 's superiority, attributed to its cell ontology-guided pre-training.

(3) Interestingly, clustering performance does not necessarily correlate directly with classification performance. For instance, while MGP+Sup outperforms Sup in classification metrics, it does not do so in clustering metrics. This observation underscores the importance to evaluate both the clustering and classification performances for cell type identification with model fine-tuning, which can make the evaluation setting more comprehensive and rigorous.

## E.3   Novel Cell Type Classification

**Method.**   In this task, we define "known cell types" $\mathcal{V}_{\mathrm{kn}} \subseteq \mathcal{V}$ as the 398 cell types from our labeled pre-training dataset (see dataset statistics Tab. 8). "Novel cell types", or "unknown cell types" $\mathcal{V}_{\mathrm{unkn}} \subseteq \mathcal{V}$, are those present only in the target downstream dataset and not observed during TFM pre-training ($\mathcal{V}_{\mathrm{unkn}} = \mathcal{V} \setminus \mathcal{V}_{\mathrm{kn}}$).

Given a new query cell $q$, we aim to classify it to one of the unknown cell types $\mathcal{V}_{\mathrm{unkn}}$. To solve this problem, we choose to first calculate representations for both the query cell sample and the unknown

cell types. And then we measure the similarity between the two representations to determine the prediction results $v_q \in \mathcal{V}_{\text{unkn}}$.

Since unknown cell types are absent from the pre-training dataset, their representations cannot be directly obtained from any TFM baselines or our model, despite its ability to learn representations for known cell types. To address this problem, we leverage the known cell types $\mathcal{V}_{\text{kn}}$ as a bridge to represent the query cells through the similarity between the cell and cell type representations produced by TFMs, and also represent the unknown cell types using the structural similarity relationships between the known and unknown ones derived from the cell ontology graph.

Specifically, our approach is illustrated in Fig. 8 and involves the following steps:

(1) **Representations for known cell types.** Although scCello inherently learns cell type representations during pre-training, most existing TFMs do not output cell type representations directly. For benchmarking, we propose a protocol to calculate known cell type representations for general TFMs. Specifically, the representation for each known cell type is calculated by averaging cell representations derived from TFMs across cells belonging to this cell type. We used cell samples from a subset (10%) of our curated pre-training dataset, because the whole 22 million dataset is too large to fit.

We denote the known cell type representations as $\{\overline{\boldsymbol{h}}_u\}_{u \in \mathcal{V}_{\text{kn}}}$, to differentiate with the notation of scCello's learned cell type representations $\{\boldsymbol{h}_u\}_{u \in \mathcal{V}_{\text{kn}}}$ introduced in Sec. 2. For fair comparison, scCello also follows this protocol to generate known cell type representations, instead of using its learned ones. Nevertheless, we emphasize scCello's capability to conduct this task alone without further accessing reference databases like our pre-training dataset.

(2) **Similarity vector for a query cell to known cell types.** We first derive the cell representations for the query cell $q$ from TFMs. Then, we estimate the similarity between the query cell $q$ and any known cell type $u \in \mathcal{V}_{\text{kn}}$ using the cosine similarity between their representations $s(q, u) = \boldsymbol{z}_q^T \overline{\boldsymbol{h}}_u$. For all known cell types, this results in a similarity vector:

$$\boldsymbol{s}(q, \boldsymbol{u}) = [d(q, u_1), d(q, u_2), \ldots, d(q, u_{|\mathcal{V}_{\text{kn}}|})] \tag{26}$$

where we define the order of vector indices as $\boldsymbol{u} = [u_1, u_2, \ldots, u_{|\mathcal{V}_{\text{kn}}|}]$ satisfying $u_1 < u_2 < \cdots < u_{|\mathcal{V}_{\text{kn}}|}$.

(3) **Similarity vector for unknown cell types to known cell types.** For each unknown cell type $v \in \mathcal{V}_{\text{unkn}}$, we estimate the similarity $\boldsymbol{s}(v, \boldsymbol{u})$ between the unknown $v$ and the known cell types $\boldsymbol{u}$. To achieve this, we leverage the cell ontology graph to calculate structural proximities as proxies. The proximities are measured using the raw PPR score $\text{PPR}(u, v), u \in \mathcal{V}_{\text{kn}}, v \in \mathcal{V}_{\text{unkn}}$, which is introduced in Sec. 2.4. Therefore, the similarity vector can be represented as:

$$\boldsymbol{s}(v, \boldsymbol{u}) = [\text{PPR}(u_1, v), \text{PPR}(u_2, v), \ldots, \text{PPR}(u_{|\mathcal{V}_{\text{kn}}|}, v)], \tag{27}$$

(4) **Align the similarity vectors for the query cell and the unknown cell types.** Intuitively, the similarity vector $\boldsymbol{s}(q, \boldsymbol{u})$ indicates a profiling for the query cell $q$, with known cell types $\boldsymbol{u}$ as a frame of reference; and the similarity vector $\boldsymbol{s}(v, \boldsymbol{u})$ conveys similar profiling for an unknown cell type $v$. Therefore, the more similar the two similarity vectors $\boldsymbol{s}(q, \boldsymbol{u})$ and $\boldsymbol{s}(v, \boldsymbol{u})$ is, the higher possibility for the query cell to be alike this unknown cell type. We derive it using Spearman Ratio [50] $\text{SpearmanR}(\cdot)$ as the similarity measure:

$$s(q, v) = \text{SpearmanR}(\boldsymbol{s}(q, \boldsymbol{u}), \boldsymbol{s}(v, \boldsymbol{u})). \tag{28}$$

Other formulas for the vector similarity function are available, like the commonly used cosine similarity (i.e., $d(q, v) = \boldsymbol{d}(q, \boldsymbol{u})^T \boldsymbol{s}(q, \boldsymbol{u})$). Our approach is not sensitive to the choice of the similarity metric. As shown in Fig. 10, using the dot product as the similarity score led to similar relative performance as in Fig. 2, where scCello generally performs better or on par with other TFMs. Therefore, we used Spearman Ratio throughout the experiments.

(5) **Select the final answer.** The unknown cell type $v^*$ with the largest distance is selected as the prediction for novel cell type classification:

$$v^* = \arg\max_{v \in \mathcal{V}_{\text{unkn}}} s(q, v) \tag{29}$$

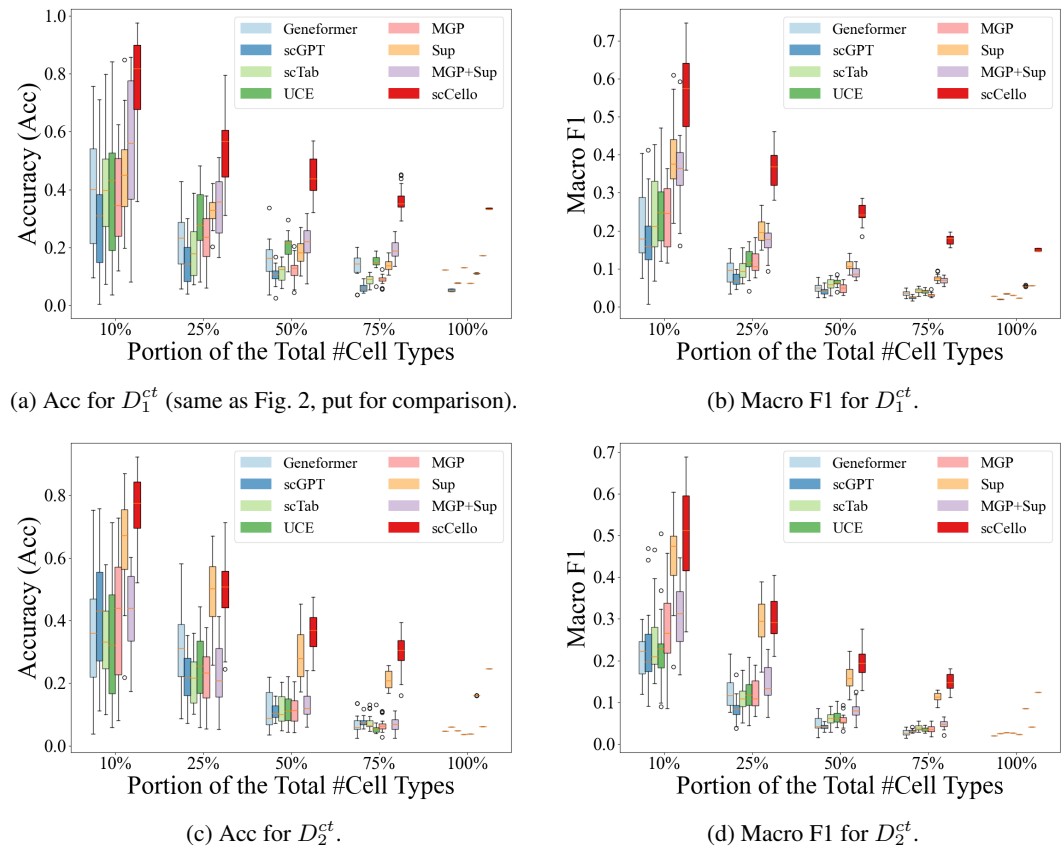

(a) Acc for $D_1^{ct}$ (same as Fig. 2, put for comparison).

(b) Macro F1 for $D_1^{ct}$.

(c) Acc for $D_2^{ct}$.

(d) Macro F1 for $D_2^{ct}$.

Figure 9: Novel cell type classification on two OOD cell type datasets $D_1^{ct}$ and $D_1^{ct}$, using the Spearman Ratio similarity measure to compare the representations of the query cells and the novel cell types (App. E.3). Two metrics Acc and Macro F1 are reported.

In real-world applications, our approach is still applicable since almost all cell types are included in the cell ontology graph. But we won't be able to know whether the newly coming query cells are from unknown cell types $\mathcal{V}_{unkn}$ or known cell types $\mathcal{V}_{kn}$. Therefore, we can expand the unknown cell type set $\mathcal{V}_{unkn}$ to all the cell type defined in the ontology graph $\mathcal{V}$, and conduct similar processes in our approach.

**Datasets.** We evaluate on OOD cell type datasets $D_1^{ct}$ and $D_2^{ct}$. The cell types in $D_1^{ct}$ and $D_2^{ct}$ are already aligned to the cell ontology graph using the ontology identifiers provided by CellxGene database, and are a subset of all the unknown cell types $\mathcal{V}_{unkn}$. We recognize that the prediction task becomes more challenging as the number of novel cell types increases. Therefore, we constrain the complete unknown cell type set to the cell types occurred in the datasets we used.

To further reflect the challenge, we created five difficulty levels, where the number of cell types spanned from 10%, 25%, 50%, 75% to 100% of the total cell type count. For example, if we use 25% cell types in the OOD cell type dataset $D_1^{ct}$ with a total 87 cell types, the unknown cell types include $(87 \times 25\% \approx 22)$ randomly selected cell types from the complete set $\{c_i | X_i \in D_1^{ct}\}$. To account for potential biases, we randomly sampled 20 distinct combinations of cell types for each difficulty level.

**Hyper-parameters.** The $\text{PPR}(\cdot)$ score is calculated using the "nx.pagerank" function with alpha hyper-parameter set to 0.9.

**Performance.** The full metrics for both accuracy and macro f1 score on the two OOD cell type datasets $D_1^{ct}$ and $D_2^{ct}$ are reported in Fig. 9. Besides plots, the numerical results are also summarized in Tab. 16 and Tab. 17 for reference.

Table 16: Novel cell type classification results on OOD cell type dataset $D_1^{ct}$.

| Method | 10% cell types | | 25% cell types | | 50% cell types | | 75% cell types | | 100% cell types | |
|--------|------|------|------|------|------|------|------|------|------|------|
| | Acc↑ | F1↑ | Acc↑ | F1↑ | Acc↑ | F1↑ | Acc↑ | F1↑ | Acc↑ | F1↑ |
| **Ontology-Agnostic TFMs** | | | | | | | | | | |
| Geneformer | 0.392 | 0.207 | 0.226 | 0.095 | 0.157 | 0.050 | 0.135 | 0.036 | 0.123 | 0.027 |
| scGPT | 0.291 | 0.178 | 0.148 | 0.072 | 0.105 | 0.041 | 0.062 | 0.024 | 0.052 | 0.020 |
| scTab | 0.380 | 0.248 | 0.191 | 0.096 | 0.114 | 0.058 | 0.088 | 0.042 | 0.077 | 0.035 |
| UCE | 0.399 | 0.253 | 0.289 | 0.120 | 0.205 | 0.064 | 0.149 | 0.040 | 0.131 | 0.030 |
| MGP | 0.361 | 0.243 | 0.233 | 0.119 | 0.125 | 0.048 | 0.089 | 0.032 | 0.076 | 0.022 |
| Sup | 0.464 | 0.389 | 0.329 | 0.200 | 0.187 | 0.109 | 0.139 | 0.075 | 0.111 | 0.055 |
| MGP+Sup | 0.556 | 0.358 | 0.341 | 0.172 | 0.217 | 0.089 | 0.193 | 0.069 | 0.172 | 0.056 |
| **Ontology-Enhanced TFMs** | | | | | | | | | | |
| **scCello** | **0.768** | **0.559** | **0.547** | **0.365** | **0.442** | **0.246** | **0.364** | **0.177** | **0.335** | **0.150** |

Table 17: Novel cell type classification results on OOD cell type dataset $D_2^{ct}$.

| Method | 10% cell types | | 25% cell types | | 50% cell types | | 75% cell types | | 100% cell types | |
|--------|------|------|------|------|------|------|------|------|------|------|
| | Acc↑ | F1↑ | Acc↑ | F1↑ | Acc↑ | F1↑ | Acc↑ | F1↑ | Acc↑ | F1↑ |
| **Ontology-Agnostic TFMs** | | | | | | | | | | |
| Geneformer | 0.367 | 0.213 | 0.310 | 0.125 | 0.107 | 0.048 | 0.069 | 0.027 | 0.047 | 0.020 |
| scGPT | 0.411 | 0.223 | 0.217 | 0.085 | 0.108 | 0.041 | 0.077 | 0.031 | 0.061 | 0.025 |
| scTab | 0.338 | 0.245 | 0.175 | 0.102 | 0.113 | 0.053 | 0.074 | 0.038 | 0.049 | 0.027 |
| UCE | 0.339 | 0.227 | 0.244 | 0.119 | 0.119 | 0.064 | 0.056 | 0.035 | 0.037 | 0.027 |
| MGP | 0.411 | 0.270 | 0.225 | 0.120 | 0.114 | 0.057 | 0.066 | 0.036 | 0.038 | 0.023 |
| Sup | 0.581 | 0.372 | 0.325 | 0.204 | 0.199 | 0.112 | 0.140 | 0.081 | 0.108 | 0.063 |
| MGP+Sup | 0.428 | 0.315 | 0.228 | 0.143 | 0.131 | 0.082 | 0.069 | 0.047 | 0.061 | 0.041 |
| **Ontology-Enhanced TFMs** | | | | | | | | | | |
| **scCello** | **0.763** | **0.500** | **0.498** | **0.304** | **0.364** | **0.196** | **0.297** | **0.149** | **0.247** | **0.124** |

## E.4  Marker Gene Prediction

**Method.**  We here explain our approach for this task in details. Given a cell's gene expression profile, we enumerate each gene and attempt to knock it out, either by replacing it with a special [MASK] token or by reducing its expression to zero. The former method is used for Geneformer, MGP, Sup, MGP+Sup, and scCello, while the latter is applied to scGPT, scTab, and UCE. By comparing the cell representations of the mutated expression and those of the original expression, we assess the impact of each gene's knockout. A greater impact suggests a higher likelihood of the gene being a marker gene. This zero-shot approach requires no further fine-tuning and is particularly useful when additional computational resources or annotated datasets for fine-tuning are unavailable.

Notably, we acknowledge the shortage of our method: for house keeping genes (*i.e.*, non-marker genes), knocking out these genes will also have large impact on the cell because the cell would die [17]. Therefore, a high impact from gene knockout does not necessarily indicate a marker gene, but rather an "important" gene. However, this issue is not critical empirically, as the number of well-documented housekeeping genes is about 400, which is small compared to the extensive gene token vocabulary of $M = 25,424$.

**Datasets.**  As introduced in Sec. 4.4, we used the datasets from GSE96583 [33] and GSE130148 [7]. One the one hand, the GSE96583 dataset $D_1^{mk}$ inherently contains five cell subsets associated with 9 cell type classes. The five cell subsets are denoted as "GSE96583_1", "GSE96583_2", "GSE96583_3", "GSE96583_4", "GSE96583_5", respectively. On the other hand, the GSE130148 dataset $D_2^{mk}$ contains 13 cell type classes. The size of these two datasets are summarized in Tab. 21, and their associated cell types are recorded in Tab. 20 for demonstration. Additionally, the ground truth cell-type-specific marker genes are originally sourced from two databases: CellMarker2 [29] and PanglaoDB [21].

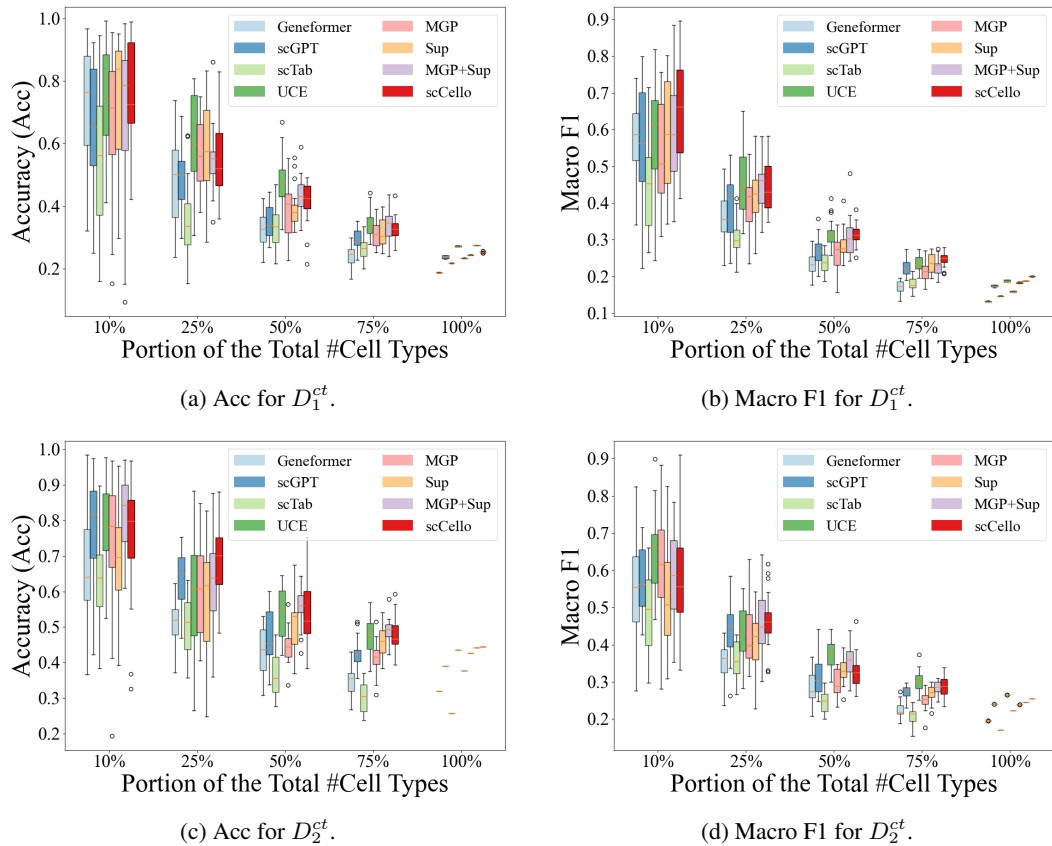

(a) Acc for $D_1^{ct}$.  (b) Macro F1 for $D_1^{ct}$.

(c) Acc for $D_2^{ct}$.  (d) Macro F1 for $D_2^{ct}$.

Figure 10: Novel cell type classification on two OOD cell type datasets $D_1^{ct}$ and $D_1^{ct}$, using the cosine similarity measure to compare the representations of the query cells and the novel cell types (App. E.3). Two metrics Acc and Macro F1 are reported.

**Performance.** In Sec. 4.4, we only reported the average performance across the 5 subsets of GSE96583 ($D_1^{mk}$) and the individual performance of GSE130148 ($D_2^{mk}$) in Tab 3. Here, we provide complete results for all five subsets in Tab. 19.

### E.5  Novel Marker Gene Prediction

**Method.** We adopted a methodology similar to our zero-shot marker gene prediction protocol (see App. E.4) for calculating differences in cell representations through in-silico gene perturbation:

(1) For each cell, we calculated the change in cell representations after removing each gene in-silico and selected the top 10% genes with the largest changes, excluding the known marker genes.

(2) For each cell type, we identified the 10 most frequent genes among the top 10% across all cells, to obtain 10 candidate novel marker genes.

(3) To ensure specificity, we removed genes present in more than one cell type.

**Datasets.** We used the same two datasets (GSE96583 and GSE130148) in Sec. 4.4 for marker gene prediction, where marker gene labels were retrieved from CellMarker2 [30] and PanglaoDB [21].

**Performance.** As a case study, we followed the above steps to find novel marker genes for two cell types:

- For cell type "CD14+ Monocytes", two genes were found: FOS and LGALS2. FOS is typically expressed in response to stress signals, cytokines, and growth factors [2]; LGALS2

Table 19: Full results for the five data subsets from GSE96583 ($D_1^{mk}$) and one dataset from GSE130148 ($D_2^{mk}$) in the marker gene prediction task (Sec. 4.4).

| Method | GSE96583 ($D_1^{mk}$) | | | | | | GSE130148 ($D_2^{mk}$) | Avg.↑ of $D_1^{mk}$ and $D_2^{mk}$ |
|---|---|---|---|---|---|---|---|---|
| | GSE96583_1 AUROC↑ | GSE96583_2 AUROC↑ | GSE96583_3 AUROC↑ | GSE96583_4 AUROC↑ | GSE96583_5 AUROC↑ | Avg.↑ | AUROC↑ | |
| **Non-TFM Methods** | | | | | | | | |
| DET | 0.725 | 0.710 | 0.737 | 0.715 | 0.717 | 0.721 | 0.683 | 0.702 |
| **Ontology-Agnostic TFMs** | | | | | | | | |
| Geneformer | 0.445 | 0.447 | 0.478 | 0.484 | 0.408 | 0.452 | 0.470 | 0.461 |
| scGPT | 0.423 | 0.387 | 0.344 | 0.385 | 0.388 | 0.385 | 0.387 | 0.386 |
| scTab | 0.666 | 0.654 | 0.689 | 0.693 | 0.660 | 0.672 | 0.727 | 0.700 |
| UCE | 0.502 | 0.499 | 0.500 | 0.499 | 0.500 | 0.500 | 0.500 | 0.500 |
| MGP | 0.572 | 0.560 | 0.606 | 0.589 | 0.567 | 0.579 | 0.629 | 0.604 |
| Sup | 0.707 | 0.697 | 0.694 | 0.699 | 0.700 | 0.699 | 0.693 | 0.696 |
| MGP+Sup | 0.734 | 0.720 | 0.739 | 0.734 | 0.724 | 0.730 | **0.730** | 0.730 |
| **Ontology-Enhanced TFMs** | | | | | | | | |
| **scCello** | **0.767** | **0.753** | **0.754** | **0.748** | **0.760** | **0.756** | **0.729** | **0.743** |

Table 20: Cell types for the two marker gene prediction datasets GSE96583 ($D_1^{mk}$) and GSE130148 ($D_2^{mk}$).

| Dataset | Cell Types |
|---|---|
| GSE96583 | "Dendritic cells", "CD8 T cells", "NK cells", "B cells", "Megakaryocytes", "FCGR3A+ Monocytes", "CD14+ Monocytes", "CD4 T cells", "Not Known" |
| GSE130148 | "Macrophages", "T cell", "NK cell", "Mast cell", "Endothelium", "Lymphatic", "Pulmonary Alveolar Type II", "Transformed epithelium', "Ciliated", "Pulmonary Alveolar Type I", "B cell", "Fibroblast", "Secretory" |

    is involved in modulating immune responses and inflammatory processes in monocytes [37]. Both are plausible marker gene candidates.

- For cell type "Megakaryocytes", five genes were found: GNG11, TUBB1, H2AC6, CAVIN2, CLU. GNG11 is confirmed as marker genes in literature, TUBB1 is a likely marker, while H2AC6, CAVIN2, and CLU require further investigation.

### E.6 Cancer Drug Response Prediction

**Method.**    In this task, we first compute cell line level representations from scRNA-seq data and drug representations for associated drugs. Both these two representations are then input into the DeepCDR framework for training. Finally, we calculate the PCC between the predicted and actual IC50 values for each drug across all cell lines and report the average performance across all tested drugs.

Specifically, for TFMs, single-cell gene expression data are inputted into each model to generate cell-specific representations for each gene. These are then aggregated into cell line-level representations through max-pooling across all genes for each dimension. Conversely, the DeepCDR method uses raw gene expressions, aggregating them directly before max-pooling. Additionally, drugs are represented as graphs and encoded using graph neural networks to obtain drug representations.

**Datasets.**    In our experiments, we utilized cell line and drug-paired data pre-processed by Deep-CDR [43], including 223 drugs and 561 cell line bulk gene expression profiles for 697 genes from 31 different cancer types. Among the dataset, 89,585 cell line-drug samples were used for training and 4,729 for testing [26].

**Hyper-parameters.**    We following scFoundation's implementation to set the parameters in the DeepCDR framework, like "-use_gexp" as True, and both "-use_mut" and "-use_methy" as False.

Table 21: The number of cell samples (#Cells) for the marker gene prediction datasets GSE96583 ($D_1^{mk}$) and GSE130148 ($D_2^{mk}$).

| Dataset | GSE96583_1 | GSE96583_2 | GSE96583_3 | GSE96583_4 | GSE96583_5 | GSE130148 |
|---------|-----------|-----------|-----------|-----------|-----------|-----------|
| #Cells | 4,246 | 3,639 | 14,619 | 14,446 | 6,145 | 10,360 |

Table 22: The correlation of the ontology structure and the pairwise similarity of known cell type representations

| Method | Spearman R↑ |
|--------|-------------|
| **Non-TFM Methods** | |
| Raw Data | 0.212 |
| Seurat | 0.316 |
| Harmony | 0.262 |
| **Ontology-Agnostic TFMs** | |
| Geneformer | 0.284 |
| scGPT | 0.037 |
| scTab | 0.209 |
| UCE | 0.285 |
| MGP | 0.275 |
| Sup | 0.229 |
| MGP+Sup | 0.238 |
| **Ontology-Enhanced TFMs** | |
| **scCello** | **0.506** |

**Performance.** Results are already reported in Tab. 4 in Sec. 4.5.

### E.7 Batch Integration

**Method.** This batch integration task aims to seamlessly integrate scRNA-seq data from different batches, which can be conducted using the same protocol as cell type clustering. After clustering, model performance is evaluated. Besides using cell type labels and clustering indices from the optimized Louvain algorithm to calculate the preservation of biological signals (NMI, ARI, ASW and AvgBio), this task also use batch labels to measure the removal of batch effects ($ASW_b$ and AvgBatch). See App. E.1 for metric calculation details.

**Datasets.** As introduced in Sec. 4.6, all datasets used in the cell type clustering task (Sec. 4.2.1) are evaluated, including one ID dataset $D^{id}$ and six OOD datasets $D_i^{cond}$ ($cond \in \{ct, ts, dn\}$, $i \in \{1, 2\}$).

**Hyper-parameters.** We use the same hyper-parameters as that in cell type clustering.

**Performance.** In Sec. 4.6, the Overall score, a weighted average of AvgBio and AvgBatch, is already reported in Fig. 3. Complete results for all metrics are included in Tab. 23 for the ID dataset $D^{id}$, Tab. 24 for the OOD cell type datasets $D_1^{ct}$ and $D_2^{ct}$, Tab. 25 for the OOD tissue datasets $D_1^{ts}$ and $D_2^{ts}$, and Tab. 26 for the OOD donor datasets $D_1^{dn}$ and $D_2^{dn}$.

### E.8 Visualization for Learned Cell Representations

We calculate known cell type representation as introduced in Sec. E.3, by averaging cell representations for each type on 10% of the pre-training data. Then we apply tSNE to project the known cell type representations to 2D space and visualize in Fig. 11. Highly correlated cell types are clustered together as expected, and dissimilar cell types are distant.

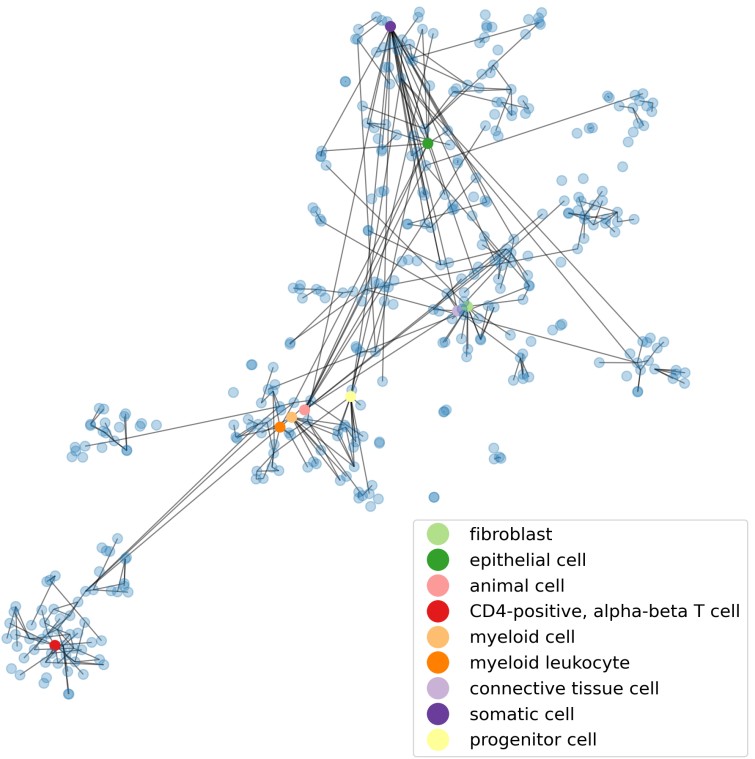

Figure 11: Visualization for learned cell representations of scCello (introduced in App. E.8). The nodes are different cell types in the pre-training dataset and the edges denote "is a subtype of" relationships in cell ontology $\mathcal{G}$. The coordinates of nodes are calculated using tSNE dimensional reduction for cell type representations derived from scCello. As expected, highly ontology-correlated cell type pairs are very close in the latent space, such as myeloid leukocyte and myeloid cell, as well as fibroblast and connective tissue cell. Meanwhile, dissimilar cell type pairs remain distant, such as CD4-positive, alpha-beta T cell and epithelial cell. The highly biologically informative representation space implies scCello's potential generalization ability to other cell-type-related downstream tasks.

We also calculate the Spearman R correlation of the pairwise similarity of known cell type representations and the ontology structure (1 for an edge between two cell types and 0 for no edge between them) in Tab. 22. As expected, scCello learned a biologically informative representation space that is much more correlated to the true ontology structure than other methods. This implies scCello's potential generalization ability to other cell-type-related downstream tasks.

We also compare the cell representation distribution for scCello with its ablated version excluding the relational alignment objective (i.e., Eqn. 4), to analyze how the ontology relational alignment loss benefits clustering performance. Following the same visualization method for Fig. 11, visualization results for the ablated model is shown in Fig. 12. Adding the relational alignment loss makes similar cell types clustered more closely and pushes dissimilar types farther apart. Therefore, relational alignment enables scCello to align better with biological intuitions, and produce more effective cell representations as evaluated by broad downstreams.

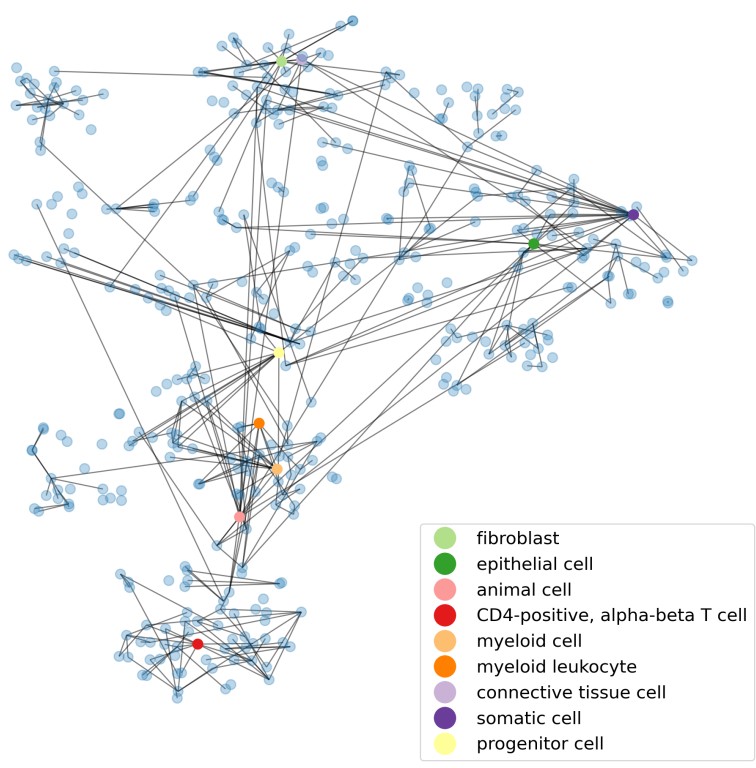

Figure 12: Visualization for learned cell representations of scCello ablation without the relational alignment objective. Compared with Fig 11, without relational alignment objective, some highly correlated cell types are not clustered well together.

Table 23: Batch integration on ID dataset $D^{id}$.

| Method | ID Unseen Data ($D^{in}$) | | | | |
|---|---|---|---|---|---|
| | ASW$_b$↑ | GraphConn↑ | AvgBatch↑ | AvgBio↑ | **Overall**↑ |
| **Non-TFM Methods** | | | | | |
| Raw Data | **0.951** | 0.806 | 0.878 | 0.419 | 0.603 |
| Seurat | 0.829 | 0.686 | 0.757 | 0.442 | 0.568 |
| Harmony | 0.824 | 0.688 | 0.756 | 0.421 | 0.555 |
| scVI | 0.880 | 0.738 | 0.809 | 0.474 | 0.608 |
| **Ontology-Agnostic TFMs** | | | | | |
| Geneformer | 0.875 | 0.676 | 0.775 | 0.432 | 0.569 |
| scGPT | 0.887 | 0.691 | 0.789 | 0.438 | 0.578 |
| scTab | 0.917 | **0.925** | **0.921** | 0.577 | **0.715** |
| UCE | 0.906 | 0.788 | 0.847 | 0.489 | 0.632 |
| MGP | 0.870 | 0.728 | 0.799 | 0.473 | 0.603 |
| Sup | 0.885 | 0.809 | 0.847 | 0.555 | 0.672 |
| MGP+Sup | 0.892 | 0.829 | 0.860 | 0.516 | 0.654 |
| **Ontology-Enhanced TFMs** | | | | | |
| **scCello** | 0.834 | 0.697 | 0.766 | **0.670** | 0.708 |

Table 24: Batch integration on OOD cell type datasets $D_1^{ct}$ and $D_2^{ct}$.

| Method | OOD CellType Data ($D_1^{ct}$) | | | | | OOD CellType Data ($D_2^{ct}$) | | | | |
|---|---|---|---|---|---|---|---|---|---|---|
| | ASW$_b$↑ | GraphConn↑ | AvgBatch↑ | AvgBio↑ | Overall↑ | ASW$_b$↑ | GraphConn↑ | AvgBatch↑ | AvgBio↑ | Overall↑ |
| **Non-TFM Methods** | | | | | | | | | | |
| Raw Data | **0.934** | 0.940 | **0.937** | 0.703 | 0.797 | **0.939** | 0.895 | **0.917** | 0.629 | 0.744 |
| Seurat | 0.831 | 0.928 | 0.880 | 0.752 | 0.803 | 0.844 | 0.932 | 0.888 | 0.737 | 0.797 |
| Harmony | 0.909 | 0.800 | 0.855 | 0.432 | 0.601 | 0.898 | 0.817 | 0.858 | 0.417 | 0.593 |
| scVI | 0.875 | **0.959** | 0.917 | 0.760 | **0.823** | 0.880 | **0.952** | 0.916 | 0.725 | 0.801 |
| **Ontology-Agnostic TFMs** | | | | | | | | | | |
| Geneformer | 0.915 | 0.907 | 0.911 | 0.689 | 0.778 | 0.915 | 0.917 | **0.916** | 0.668 | 0.767 |
| scGPT | 0.903 | 0.913 | 0.908 | 0.707 | 0.787 | 0.896 | 0.927 | **0.912** | 0.720 | 0.797 |
| scTab | 0.908 | 0.904 | 0.906 | 0.759 | 0.818 | 0.910 | 0.905 | 0.908 | 0.726 | 0.799 |
| UCE | 0.867 | 0.947 | 0.907 | **0.772** | **0.826** | 0.854 | 0.946 | 0.900 | 0.741 | 0.805 |
| MGP | 0.894 | 0.903 | 0.899 | 0.714 | 0.788 | 0.925 | 0.580 | 0.753 | 0.740 | 0.745 |
| Sup | 0.879 | 0.944 | 0.912 | **0.767** | **0.825** | 0.879 | 0.914 | 0.897 | 0.775 | **0.824** |
| MGP+Sup | 0.885 | 0.946 | 0.916 | 0.758 | **0.821** | 0.885 | 0.925 | 0.905 | 0.764 | **0.820** |
| **Ontology-Enhanced TFMs** | | | | | | | | | | |
| **scCello** | 0.877 | 0.911 | 0.894 | **0.769** | 0.819 | 0.858 | 0.884 | 0.871 | **0.786** | **0.820** |

Table 25: Batch integration on OOD tissue datasets $D_1^{ts}$ and $D_2^{ts}$.

| Method | OOD Tissue Data ($D_1^{ts}$) | | | | | OOD Tissue Data ($D_2^{ts}$) | | | | |
|---|---|---|---|---|---|---|---|---|---|---|
| | ASW$_b$↑ | GraphConn↑ | AvgBatch↑ | AvgBio↑ | Overall↑ | ASW$_b$↑ | GraphConn↑ | AvgBatch↑ | AvgBio↑ | Overall↑ |
| **Non-TFM Methods** | | | | | | | | | | |
| Raw Data | **0.941** | 0.792 | 0.867 | 0.540 | 0.671 | **0.946** | 0.862 | 0.904 | 0.631 | 0.740 |
| Seurat | 0.865 | 0.830 | 0.847 | 0.587 | 0.691 | 0.867 | 0.841 | 0.854 | 0.636 | 0.723 |
| Harmony | 0.905 | 0.755 | 0.830 | 0.462 | 0.609 | 0.908 | 0.744 | 0.826 | 0.515 | 0.639 |
| scVI | 0.901 | 0.861 | 0.881 | 0.577 | 0.699 | 0.910 | 0.881 | 0.896 | 0.634 | 0.739 |
| **Ontology-Agnostic TFMs** | | | | | | | | | | |
| Geneformer | 0.925 | 0.804 | 0.865 | 0.539 | 0.669 | 0.924 | 0.835 | 0.880 | 0.597 | 0.710 |
| scGPT | 0.916 | 0.776 | 0.846 | 0.544 | 0.665 | 0.920 | 0.826 | 0.873 | 0.627 | 0.725 |
| scTab | 0.916 | 0.872 | 0.894 | 0.515 | 0.667 | 0.917 | 0.874 | 0.896 | 0.657 | 0.753 |
| UCE | 0.905 | 0.864 | 0.885 | 0.598 | 0.713 | 0.911 | 0.879 | 0.895 | 0.670 | 0.760 |
| MGP | 0.887 | 0.887 | 0.887 | 0.576 | 0.700 | 0.901 | 0.815 | 0.858 | 0.628 | 0.720 |
| Sup | 0.903 | 0.932 | **0.918** | 0.605 | **0.730** | 0.899 | 0.911 | **0.905** | 0.680 | **0.770** |
| MGP+Sup | 0.900 | **0.941** | 0.921 | **0.610** | **0.734** | 0.898 | **0.922** | 0.910 | 0.672 | **0.767** |
| **Ontology-Enhanced TFMs** | | | | | | | | | | |
| **scCello** | 0.868 | 0.841 | 0.855 | **0.612** | 0.709 | 0.884 | 0.819 | 0.852 | **0.705** | 0.764 |

Table 26: Batch integration on OOD donor datasets $D_1^{dn}$ and $D_2^{dn}$.

| Method | OOD Donor Data ($D_1^{dn}$) | | | | | OOD Donor Data ($D_2^{dn}$) | | | | |
|---|---|---|---|---|---|---|---|---|---|---|
| | ASW$_b$↑ | GraphConn↑ | AvgBatch↑ | AvgBio↑ | Overall↑ | ASW$_b$↑ | GraphConn↑ | AvgBatch↑ | AvgBio↑ | Overall↑ |
| **Non-TFM Methods** | | | | | | | | | | |
| Raw Data | **0.945** | 0.785 | 0.865 | 0.458 | 0.621 | **0.946** | 0.787 | 0.867 | 0.460 | 0.623 |
| Seurat | 0.875 | 0.759 | 0.817 | 0.466 | 0.606 | 0.876 | 0.771 | 0.824 | 0.489 | 0.623 |
| Harmony | 0.893 | 0.618 | 0.756 | 0.456 | 0.576 | 0.891 | 0.655 | 0.773 | 0.474 | 0.594 |
| scVI | 0.914 | 0.831 | 0.872 | 0.478 | 0.636 | 0.909 | 0.837 | 0.873 | 0.502 | 0.650 |
| **Ontology-Agnostic TFMs** | | | | | | | | | | |
| Geneformer | 0.921 | 0.763 | 0.842 | 0.468 | 0.618 | 0.919 | 0.768 | 0.844 | 0.482 | 0.627 |
| scGPT | 0.920 | 0.757 | 0.839 | 0.456 | 0.609 | 0.920 | 0.763 | 0.842 | 0.477 | 0.623 |
| scTab | / | / | / | OOM | OOM | / | / | / | OOM | OOM |
| UCE | 0.904 | 0.665 | 0.784 | 0.485 | 0.605 | 0.907 | 0.558 | 0.733 | 0.506 | 0.597 |
| MGP | 0.910 | 0.824 | 0.867 | 0.488 | 0.640 | 0.906 | 0.814 | 0.860 | 0.518 | 0.655 |
| Sup | 0.909 | 0.877 | 0.893 | 0.552 | 0.688 | 0.902 | 0.857 | 0.880 | 0.573 | 0.696 |
| MGP+Sup | 0.910 | **0.888** | 0.899 | 0.553 | **0.691** | 0.903 | **0.869** | **0.886** | 0.570 | 0.696 |
| **Ontology-Enhanced TFMs** | | | | | | | | | | |
| **scCello** | 0.845 | 0.805 | 0.825 | **0.608** | **0.695** | 0.849 | 0.802 | 0.826 | **0.643** | **0.716** |

