# OpenReview forum: "Cell ontology guided transcriptome foundation model"
_NeurIPS.cc/2024/Conference — NeurIPS 2024 spotlight_

### Official Review · Reviewer_hRwa · 2024-07-06

**Soundness:** 4
**Presentation:** 4
**Contribution:** 3
**Rating:** 7
**Confidence:** 4

**Summary:**

This paper introduces scCello, a Transcriptome Foundation Model (TFM) which incorporates prior information from cell ontology graphs denoting the relationships between different cell types in order to guide cell representations. The pretraining objective in the scCello framework encapsulates masked gene expression prediction, a supervised contrastive objective for aligning cell representations to cell ontology terms, and relational alignment of similar cell types, which helps guide representation learning during pretraining to reflect  taxonomic relationships between cells. The foundation model, scCello, is pretrained on 22 million cells from the CellxGene public data repository, and demonstrates improved performance over other single-cell foundation models in downstream tasks, including cell type clustering, cell type identification, marker gene prediction, and batch integration.

**Strengths:**

This work presents a novel foundation model for single-cell transcriptomics, scCello, that incorporates prior information about cell ontology in order to guide representation learning over a large pretraining corpus. The methodology and motivation for the paper is clear, and the work represents a significant step towards training foundational models that incorporate prior information, which subsequent works may build and improve upon.

For pretraining, the authors introduce a multi-component loss function, encapsulating: (i) masked gene expression prediction, (ii) a supervised contrastive objective to align representations to of cells with similar cell ontology terms, and (iii) a relational alignment objective which enforces cell representations to follow the structural similarity of their cell types in the cell ontology graph. For quantifying cell type similarity, the Personalized PageRank (PPR) algorithm is used to derive structural similarity of pairwise cell types in the cell ontology graph, which is then used to guide representation learning.

The single-cell RNA seuqencing data preprocessing methodology seems sound, and follows common techniques for data preprocessing used by other foundation models. The authors also include a hyperparameter and compute resource comparison of scCello against other established single-cell foundational models - Geneformer, scGPT, scTAB, UCE.

Overall, the paper is well-written, motivated well, and clearly explained. Adequate details are provided for the loss objectives and methodology, and additional dataset acquisition and preprocessing details are provided in the appendix.

**Weaknesses:**

The authors mention that cell type ontology identifiers were obtained for the 22 million cell pretraining dataset from the CellxGene database, to enable mapping between individual cells and cell ontology. While this allows for additional priors from cell ontology graphs to be used during pretraining of scCello, it also necessitates labeled data during pretraining, which may limit the scalability of scCello in terms of available pretraining data compared to other methods which do not require annotated pretraining single-cell data.

**Questions:**

For the overall pretraining objective in equation 5, more details could be added about how the four loss terms are weighted relative to one another. Are their coefficients to weigh the different loss terms? If not, do the magnitudes of the different loss terms differ significantly?

The authors quantify ontology relationships in the cell ontology graph by using structural similarity, and use the Personalized PageRank (PPR) algorithm to estimate pairwise node similarities for cell types. Examples of cell types associated with varying levels of structural similarity are also presented in Table 7. It would be helpful if the authors provided a motivation for going with a structural similarity measure such as PPR score, rather than a simpler metric such as the distance of two nodes in the cell ontology graph.

**Limitations:**

The authors have adequately addressed the limitations and potential negative impacts of their work, and agree to do a code implementation release upon acceptance of the paper.

---

> ### Author Rebuttal · Authors · 2024-08-06
>
> Thanks for positive reviews and invaluable suggestions  We respond to your questions as below:
>
> &nbsp;
> > **W1: Although cell type ontology labels allow the use of cellular ontology structural priors to guide TFM pre-training, the need for labeled data may limit scCello's scalability in fully utilizing unannotated scRNA-seq data.**
>
> Thanks for bringing this up. We would like to argue that **scCello can also leverage unannotated scRNA-seq data**, by only using its masked gene prediction objective, and neglecting the intra- and inter- cell-level losses that require cell type labels. This reduces scCello to Geneformer-like TFMs. Therefore, scCello can be trained on partially labeled scRNA-seq data. We will explore this scenario in future work.
>
> &nbsp;
> > **Q1: How are the four loss terms in Eqn. 5 weighted relative to one another? And do the magnitude of these loss terms differ significantly?**
>
> The weights between the four loss terms in Eqn. 5 **are simply all ones.** And the magnitude of these loss terms **do not differ significantly.**
>
> For visualization, **four training loss figures are included in Figure R1 in the PDF file attached to the global rebuttal response.** Specifically, the training process takes 40k steps as detailed in Sec. 4.1, and the magnitudes of losses are roughly as follows:
> - The masked gene prediction loss $\mathcal{L}_{\textrm{MGP}}$ starts from 19.55 (1 step), achieves 8.09 (2k steps), and ends at 3.57 (40k steps)
> - The intra-cellular ontology coherence loss $\mathcal{L}_{\textrm{Intra}}$ starts from 13.14 (1 step), achieves 1.29 (2k steps), and ends at 0.39 (40k steps).
> - The inter-cellular relational alignment loss $\mathcal{L}_{\textrm{Inter}}$ starts from
> 1.99 (1 step), achieves 0.57 (2k steps), and ends at 0.26 (40k steps).
> - The regularization term starts from 0.51 (1 step), achieves 0.03 (2k steps), and ends at 0.02 (40k steps).
>
> In summary, all four losses weighted with all ones do not differ significantly in magnitudes, and can be optimized effectively throughout the learning process as shown in Figure R1.
>
> &nbsp;
> > **Q2: Can authors provide a motivation for going with a structural similarity measure such as PPR score, rather than a simpler metric such as node distance.**
>
> The PPR metric considers structural topology between node pairs. Intuitively, for a target node and a query node, **PPR performs graph propagations to gather information from multiple paths between this pair of nodes via random walking** (see App. A for algorithm details). In contrast,  simple metrics like **pairwise distance only consider the shortest paths.** Therefore, PPR inherently is more informative with a stronger capability to capture subgraph structures, and may be more generalizable compared to simpler metric like node distance.

---

> > ### Comment · Reviewer_hRwa · 2024-08-11
> >
> > Thank you for the detailed feedback and for the training loss figure provided in Figure R1 of the PDF document. My main questions regarding labeled data and structural similarity are answered, and I would like to keep my positive score.

---

> > > ### Author Response · Authors · 2024-08-14
> > >
> > > Dear Reviewer,
> > >
> > > Thanks for your strong support on our project! We're glad that the rebuttal results helped. And we highly appreciate your time, efforts and the positive rating!
> > >
> > > Best,
> > >
> > > Authors

---

### Official Review · Reviewer_E5ju · 2024-07-12

**Soundness:** 3
**Presentation:** 3
**Contribution:** 3
**Rating:** 7
**Confidence:** 3

**Summary:**

The authors introduced a transcriptome foundation model (TFM)  named scCello to resolve the current problem of most of the TFMs they treat cells as independent samples and ignore the taxonomic relationships between cell types. By integrating cell ontology information as well as incorporating three key objectives in the pretraining framework: masked gene prediction, cell type coherence loss, and ontology alignment loss, the model can learn gene coexpression patterns, cell type-specific representations, and structural relationships between cell types. The authors performed a series of experiments to show that the model can predict the unseen cell types, integrate datasets across diverse batches, cluster cell types, predict the cancer response to the drug, and predict the marker genes.

**Strengths:**

1. The manuscript is well-written and well-structured, and it is easy to read.
2. The authors effectively incorporate cell type ontology information at the pertaining stage, demonstrating in the comprehensive experiment results that this ontology information enhances the model's ability to learn cell type taxonomic relationships, thereby producing more biologically meaningful cell representations.

**Weaknesses:**

1. In the zero-shot cell type clustering experiment, it is not clear that the authors run the Louvain algorithm with default settings or the optimal setting on Seurat, Harmony, and scVI.
2. In the marker gene prediction part, what is the difference between predicted marker genes based on scCello and the traditional marker genes based on the differential expression level?
3. There is a paper (A Deep Dive into Single-Cell RNA Sequencing Foundation Models) indicating that the L1 logistic regression model achieves decent performance in cell type annotation task. Could the authors add the L1 logistic regression model for comparison in Table 2?

**Questions:**

1. How did the method achieve a high score in the batch integration? In the loss function, there is no specific loss for batch correction. Could the authors give some explanations for this question?
2. Can the model predict the novel marker genes?

**Limitations:**

The authors have addressed all limitations.

---

> ### Author Rebuttal · Authors · 2024-08-06
>
> Thanks for your positive review and insightful comments! We respond to your concerns as below:
>
> &nbsp;
> > **W1: In zero-shot cell type clustering, do authors use the optimal setting or the default setting for the Louvain Algorithm for Seurat, Harmony and scVI?**
>
> In zero-shot cell type clustering, we **used the optimal setting for the resolution** hyper-parameter in the Louvain algorithm across both TFM and non-TFM baselines, including Seurat, Harmony, and scVI. Notably, for all tasks involving the Louvain algorithm to calculate AvgBio or AvgBatch, we used the optimal resolution.
>
> &nbsp;
> > **W2: In marker gene prediction, how does the performance of scCello compare to the traditional marker genes prediction method based on the differential expression level?**
>
> Table D: Extended results for Differential Expression Tests (DET) for marker gene prediction on the two datasets $D^{mk}_1$ and $D^{mk}_2$ used in Sec. 4.4. AUROC is reported.
>
> | **Method** | **$D^{mk}_1$** | **$D^{mk}_2$** | **Avg.** |
> |--- |---  |--- |--- |
> | scCello |0.756 | 0.729 | 0.743|
> | DET | 0.721 | 0.683 | 0.702 |
>
> &nbsp;
>
> As suggested, we add the Differential Expression Tests (DET)[b] as a traditional baseline, implemented with Seurat [c]. DET determines marker genes by comparing gene expression among cells of one cell type against cells from other cell types.
>
> For the metric threshold, we follow conventions [c] to set `p_val_adj` < 1 and `avg_log2FC` > 0, where `p_val_adj` refers to the adjusted p-value, and `avg_log2FC` refers to the log fold-change of the average expression between the two groups.
>
> **As shown in the Table D above, scCello outperforms DET.**
>
> [b] Soneson et al. "Bias, robustness and scalability in single-cell differential expression analysis." Nature methods 2018
>
> [c] Hao et al. "Dictionary learning for integrative, multimodal and scalable single-cell analysis." Nature biotechnology 2024
>
> &nbsp;
> > **W3: For the fine-tuning setting of cell type identification, could authors add the L1 Logistic Regression (L1-LR) model for baseline comparison?**
>
> Table E: Extended results for L1-LR for cell type identification under fine-tuning setting. The classification (Clf.) and clustering (Clst.) performances on the ID dataset $D^{id}$ are reported. The clustering performance for the L1-LR model is not reported because L1-LR is a classification method and there are no internal cell representations for clustering.
>
> | **Method** | **Clf. Acc** | **Clf. Macro F1** | **Clst. AvgBio** |
> |--- |---  |--- |--- |
> | scCello | 0.867 | 0.511 | 0.694|
> | L1-LR| 0.747 | 0.491 | / |
>
> &nbsp;
>
> **Yes. We add the L1-LR model as a baseline**, following the implementation from the above suggested paper [d]. **As shown in the Table E above, scCello outperforms L1-LR.**
>
> Interestingly, compared to Tab. 2 in paper, L1-LR indeed outperforms all other TFM baselines on Macro-F1, which agrees with the previous finding in [d].
>
> [d] Boiarsky et al. "A deep dive into single-cell RNA sequencing foundation models." bioRxiv 2023
>
> &nbsp;
> > **Q1: How did scCello achieve a high score in the batch integration, without explicitly considering batch correction objectives?**
>
> The primary reason is that **scCello injects cell type lineage structural priors into its TFM pre-training, enhancing model generalization and robustness.** This allows scCello to better capture biological signals in scRNA-seq data, improving cell representation learning and mitigating batch biases.
>
> Additionally, scCello employs Geneformer’s [e] **Rank Value Encoding** approach (see App. C) to tokenize scRNA-seq data into ordered tokens. This rank-based method offers better robustness against technical artifacts compared to using raw numerical expressions, which can vary significantly across different assays.
> Lastly, we focused on scRNA-seq data from 10x assays, following scTab’s [f] preprocessing steps. **This minimized technical biases in pre-training.** We plan to explore training on more heterogeneous data from diverse scRNA-seq platforms in future work.
>
> [e] Theodoris et al. "Transfer learning enables predictions in network biology." Nature 618.7965,2023
>
> [f] Fischer et al. "Scaling cross-tissue single-cell annotation models." bioRxiv 2023
>
> &nbsp;
> > **Q2: Can the model predict the novel marker genes?**
>
> **Yes, we can use the following approach**, where the calculation of cell representation difference via in-silico gene perturbation is similar to the protocol used in our zero-shot marker gene prediction (see App. D.4).
>
> 1. We used the same two datasets (GSE96583 and GSE130148) in Sec. 4.4 for marker gene prediction, where marker gene labels were retrieved from CellMarker2 and PanglaoDB.
> 2. For each cell, we calculated the change in cell representations after removing each gene in-silico and selected the top 10% genes with the largest changes, excluding the known marker genes.
> 3. For each cell type, we identified the 10 most frequent genes among the top 10% across all cells, to obtain 10 candidate novel marker genes.
> 4. To ensure specificity, we removed genes present in more than one cell type.
>
>
> As a case study, we followed the above steps to find novel marker genes for two cell types:
> - For cell type “CD14+ Monocytes”, two genes were found: FOS and LGALS2. FOS is typically expressed in response to stress signals, cytokines, and growth factors [g]; LGALS2 is involved in modulating immune responses and inflammatory processes in monocytes [h]. Both are plausible marker gene candidates.
> - For cell type “Megakaryocytes”, five genes were found: GNG11, TUBB1, H2AC6, CAVIN2, CLU. GNG11 is confirmed as marker genes in literature, TUBB1 is a likely marker, while H2AC6, CAVIN2, and CLU require further investigation.
>
> [g] Angel, et al. The role of Jun, Fos and the AP-1 complex in cell-proliferation and transformation. Biochimica et Biophysica Acta (BBA)-Reviews on Cancer, 1072(2-3), 129-157.
>
> [h] Kim, et al. Glycobiology of innate immunology. Berlin/Heidelberg, Germany: Springer, 2022.

---

> > ### Comment · Reviewer_E5ju · 2024-08-11
> >
> > The reviewer has addressed all my concerns. I will keep my score unchanged.

---

> > > ### Author Response · Authors · 2024-08-14
> > >
> > > Dear Reviewer,
> > >
> > > Thanks for your strong support on our project and the insightful comments! We would additional experimental results and the new interesting downstream task "novel marker prediction" in our next revised version, to make the paper content more enriched and comprehensive.
> > >
> > > Best,
> > >
> > > Authors

---

### Official Review · Reviewer_bUQe · 2024-07-12

**Soundness:** 3
**Presentation:** 3
**Contribution:** 2
**Rating:** 5
**Confidence:** 4

**Summary:**

This paper presents scCello, a single-cell, Cell-ontology guided Transcriptome Foundation Model (TFM) that leverages cell-type relationships from cell ontology graphs to enhance cell representation learning. scCello incorporates three levels of objectives during pre-training: masked gene prediction, cell-type coherence, and ontology alignment. This approach improves the model's generalization and transferability capabilities, leading to superior performance in tasks such as cell type identification, novel cell type classification, marker gene prediction, and cancer drug response prediction. scCello is also robust to batch effects and demonstrates high parameter efficiency compared to existing TFMs.

**Strengths:**

1.Integrating cell ontology graphs into TFM pre-training: This is the most innovative aspect of the paper. scCello improves model understanding of biological relationships between cells by incorporating cell-type relationships from cell ontology graphs into TFM pre-training, enhancing the model's generalization capability and transferability.
2.Introducing a multi-level objective function: scCello employs a multi-level objective function that encompasses gene-level masked gene prediction, intra-cellular cell-type coherence, and inter-cellular ontology alignment. This multi-level approach enables the model to learn complex relationships between genes, cells, and cell types, leading to more precise and robust cell representations.
3.Validating the model's effectiveness and advantages: The paper presents a comprehensive set of experiments demonstrating the effectiveness and superiority of the scCello model in tasks such as cell type identification, novel cell type classification, marker gene prediction, and cancer drug response prediction. Additionally, the paper highlights scCello's robustness in handling batch effects and its parameter efficiency, suggesting the model's strong potential for real-world applications.

**Weaknesses:**

1.Lack of Theoretical Justification: The paper heavily relies on empirical results to showcase scCello's effectiveness. However, it lacks a rigorous theoretical analysis of the proposed approach. A deeper theoretical understanding of the objective functions, their impact on learning, and how they contribute to generalization capability would significantly strengthen the paper.
2.Limited Comparative Analysis: The paper primarily focuses on comparing scCello with existing TFMs but doesn't thoroughly analyze its performance against methods specifically designed for downstream tasks like cell type identification, marker gene prediction, and cancer drug response prediction. A more robust comparison with specialized methods would better illustrate scCello's strengths and highlight its relative advantages.
3.The paper's current model size might be insufficient to fully capture the complexities of single-cell transcriptomic data. This limitation could hinder the model's ability to achieve optimal performance on tasks requiring deep understanding of complex biological processes, especially when dealing with large and diverse datasets.

**Questions:**

1.Can the authors elaborate on the theoretical foundations underlying the effectiveness of integrating cell ontology into TFM pre-training? Specifically, how do the proposed objective functions contribute to learning more informative and robust cell representations?
2.Can the authors provide a more comprehensive comparison of scCello's performance against specialized methods designed for downstream tasks like cell type identification, marker gene prediction, and cancer drug response prediction?

**Limitations:**

1.The cell ontology is constantly evolving, and the paper acknowledges that scCello currently requires retraining the entire model for any updates. This limitation hinders the model's adaptability and its ability to keep up with the latest knowledge in cell biology.
2.The paper states that they aim to scale up the model size in future work. This indicates that the current model size might not be sufficient to fully capture the complexities of single-cell transcriptomic data, potentially limiting its performance on certain tasks or datasets.

---

> ### Author Rebuttal · Authors · 2024-08-06
>
> Thanks for your comments! We respond to your concerns as below:
>
> &nbsp;
> > **W1: This paper lacks of theoretical justification for deeper understanding of objective’s impact on learning and model’s generalization capability**.
>
> Our paper focuses on developing a new algorithm to improve TFM pre-training, which facilitates practical cell-related and gene-related downstream tasks. To show the effectiveness of scCello, we focus on empirical evidence on real-world data. While we appreciate the **theoretical justification for model generalization capability like analysis of error bound, it is outside the scope of our current study.** We would like to leave that as future work.
>
> &nbsp;
> > **W2 & Q2: This paper lacks of non-TFM baselines specialized for downstream tasks like cell type identification, marker gene prediction, and cancer drug response prediction**.
>
> Table B: Extended results for scANVI and L1-LR for cell type identification under the fine-tuning setting. The classification (Clf.) and clustering (Clst.) performances on the ID dataset $D^{id}$ are reported. The clustering performance for L1-LR is not reported because L1-LR is a classification method and there are no internal cell representations for clustering.
> | **Method** | **Clf. Acc** | **Clf. Macro F1** | **Clst. AvgBio** |
> |--- |---  |--- |--- |
> | scCello | 0.867 | 0.511 | 0.694|
> | scANVI |0.382 | 0.024 | 0.472|
> | L1-LR| 0.747 | 0.491 | / |
> &nbsp;
>
> Table C: Extended results for DET for marker gene prediction on the two datasets $D^{mk}_1$ and $D^{mk}_2$ used in Sec. 4.4. AUROC is reported
> | **Method** | **$D^{mk}_1$** | **$D^{mk}_2$** | **Avg.** |
> |--- |---  |--- |--- |
> | scCello |0.756 | 0.729 | 0.743|
> | DET | 0.721 | 0.683 | 0.702 |
>
> &nbsp;
>
> Following the reviewer’s suggestion, we specify existing specialized baselines and **add three more baselines**:
>
> - **For the zero-shot setting of cell type identification**, we included non-TFM specialized methods like “Raw Data”, “Seurat”, “Harmony” and “scVI” in Sec. 4.2.1
> - **For the fine-tuning setting of cell type identification**, we add two more baselines: scANVI[a] implemented with scvi library, and L1-regularized Logistic Regression (L1-LR)[b] implemented with Scikit-Learn library. **As shown in Table B above, scCello outperforms both scANVI and L1-LR**.
> - **For marker gene prediction**, we add one traditional method Differential Expression Tests (DET)[c] to identify cell-type-specific marker genes, following Seurat’s [d] implementation. DET determines marker genes by comparing gene expression among cells of one cell type against cells from other cell types. **As shown in Table C above, scCello outperforms DET**.
> - **For cancer drug response prediction**, we included a specialized non-TFM method DeepCDR[e] in Sec. 4.5, which was the state-of-the-art traditional method.
>
> [a] Xu, et al. "Probabilistic harmonization and annotation of single‐cell transcriptomics data with deep generative models." Molecular systems biology 2021
>
> [b] Vidaurre et al. "A survey of L1 regression." International Statistical Review 2013
>
> [c] Soneson et al. "Bias, robustness and scalability in single-cell differential expression analysis." Nature methods 2018
>
> [d] Hao et al. "Dictionary learning for integrative, multimodal and scalable single-cell analysis." Nature biotechnology 2024
>
> [e] Liu, Qiao et al. "DeepCDR: a hybrid graph convolutional network for predicting cancer drug response." Bioinformatics 2020
>
> &nbsp;
> > **W3 & L2: scCello’s current model size could be insufficient to handle the large and complex scRNA-seq, and achieve the optimal performance on downstreams**
>
>
> As discussed in Sec. 5, we leave the scaling law study of scCello as future work due to limited computational resources and because scaling is not the focus of this paper. Our primary focus is on developing new algorithms to improve TFM pre-training strategies.
>
> &nbsp;
> > **Q1: Analyze the effectiveness of each component of scCello’s objectives to integrate cell ontology into TFM pre-training for better cell representation learning**
>
> **The general effectiveness** arises in **injecting prior structural knowledge** (i.e., the ontology graph) into the representation learning process. Knowledge injection can advance model generalization, as shown in previous studies[f,g].
>
> **For each loss component**, (1) the marker gene prediction loss captures dynamic gene co-expression patterns, to enrich the understanding of gene interactions; (2) the intra-cellular ontology coherence loss encourages cell representations of the same cell type to aggregate, prompting consistency between cells and their types; (3) the inter-cellular relational alignment loss guides the cell representation learning by injecting the cell-type lineage relationships derived from the cell ontology graph.
>
> [f] Martino et al. "Knowledge injection to counter large language model (LLM) hallucination." European Semantic Web Conference. Cham: Springer Nature Switzerland 2023
>
> [g] Ovadia et al. "Fine-tuning or retrieval? comparing knowledge injection in llms." arXiv preprint arXiv:2312.05934,2023
>
> &nbsp;
> > **L1: scCello currently requires retraining the entire model for any updates from the constantly evolving cell ontology, which hinders its adaptability**
>
> As discussed in Sec. 5, we leave the support of dynamic growing ontology as future work. Notably, while Cell Ontology updates a few times a year[h], most of the ontology remains stable. Therefore, **we can fine-tune scCello with relatively much less computation costs**, by using recent advances in dynamic graph representation learning[i]. These methods, which handle evolving nodes, attributes, and edges over time, may effectively incorporate new changes in ontology graphs.
>
> [h] Diehl et al. "The Cell Ontology 2016: enhanced content, modularization, and ontology interoperability." Journal of biomedical semantics 2016
>
> [i] Kazemi et al. "Representation learning for dynamic graphs: A survey." Journal of Machine Learning Research 2020

---

> > ### Comment · Reviewer_bUQe · 2024-08-12
> >
> > Thanks for addressing some of my questions directly and some as future work. Taking all factors into consideration, I will keep my score unchanged.

---

> > > ### Author Response · Authors · 2024-08-14
> > >
> > > Dear Reviewer,
> > >
> > > Thanks for the thorough reviewing and suggestions on our paper! We would incorporate the added experimental results and discussions in our next revised version based on the suggestion.
> > >
> > > Best,
> > >
> > > Authors

---

### Official Review · Reviewer_BhVy · 2024-07-13

**Soundness:** 4
**Presentation:** 4
**Contribution:** 3
**Rating:** 7
**Confidence:** 4

**Summary:**

This paper introduces a new transcriptome foundation model (scCello) to generate cellular representations from single-cell RNA-seq data. The key contribution is the integration of known cell type labels (previously annotated by CellxGene submitters) within two novel objectives. First, the authors introduce a cell-type coherence loss to minimize the distance between a cell's representation and the learned representation of its associated cell type. Second, the authors introduce an ontology relational alignment loss to ensure that the similarity between the representations of two cells matches the similarity of their corresponding cell types in an ontology graph. The authors benchmark their model against previously pre-trained transcriptome foundation models and traditional (non-foundation) models. They evaluate performance on tasks including zero-shot cell clustering, fine-tuned cell type classification, and marker gene discovery and find that scCello is state of the art on almost all tasks.

**Strengths:**

- *Conceptually simple but effective*: Utilizing cell type annotations, which are almost always available in published scRNA-seq datasets, to enhance the model's representations is smart and relatively straightforward. Other methods have proposed incorporating cell type labels before, but a cell type classification loss that treats each cell type as independent has shortcomings. The ontology relational alignment loss presented here which uses pre-existing ontology graphs to determine the similarity between cell types seems intuitively effective and empirically turns out to be too.

- *Meaningful task selection*: The authors do a good job benchmarking their model on meaningful tasks. For example, they evaluate clustering performance not just on heldout IID data, but also in unobserved cell types, tissues, and donors. The model's improved performance on novel cell type classification and marker gene prediction suggests that it has captured biologically meaningful information.

- *Appropriate baselines*: The authors do not just compare to existing transcriptome foundation models, which have been heavily criticized for not being competitive with traditional methods.

- *Algorithm for novel cell type classification*: Their method of comparing the similarity vector between a cell and prototype representations of each cell type to the similarity vector between a cell type and all other cell types, as derived from the ontology graph, is clever.

**Weaknesses:**

- *Lacking analysis on ontology relational alignment loss*: While it seems intuitive that the ontology relational alignment loss is beneficial, the significant performance improvements are unexpected and probably warrant further analysis. For example, understanding which cell types show improved clustering performance could help provide some intuition.

- *Missing some baselines*: Since the model assumes that cell type annotations are available, it should be compared against traditional methods that utilize these annotations (e.g. scANVI).

- *Limited to 10x data*: The batch correction results, as is, are impressive, but it would be interesting to see if this method could handle data from different sequencing technologies.

**Questions:**

1. The regularization loss (Eq. 2) was proposed to prevent class collapse. But it's unclear why this would help. In fact, it seems to be encouraging all samples of cell type c_{i} to have the same embedding: Linear(h_{c_{i}}).
2. How did you train scVI for Table 1? Did you train it on your pre-training dataset, in-distribution dataset, and out-of-distribution datasets jointly? If not, for a fair comparison, it seems like you should.
3. To avoid picking up housekeeping genes in the marker gene analysis, could you see if knocking out housekeeping genes leads to a representation change that differs from that induced by knocking out marker genes. I imagine knocking out housekeeping genes might shift the cells to a previously uncharted part of the latent space (dead cells), whereas knocking out marker genes might shift the cells to a different cell type.

**Limitations:**

The authors are forthcoming about the limitations of their model.

---

> ### Author Rebuttal · Authors · 2024-08-06
>
> Thanks for the positive review and constructive comments, which we carefully address below:
>
> &nbsp;
>
> > **W1: Provide example for cell types to analyze how the ontology relational alignment loss benefits clustering performance**
>
> To compare scCello with its ablation excluding the relational alignment term, we visualize their cell representation distributions. Following App. D.7, we calculate prototype representations by averaging cell representations for each type on 10% of the pre-training data, and visualize them via tSNE. **The tSNE figures are in Figure R2 in the PDF file attached to the global rebuttal response.**
>
> As shown in Figure R2, adding the relational alignment loss makes similar cell types clustered more closely and pushes dissimilar types farther apart. Therefore, relational alignment enables scCello to align better with biological intuitions, and produce more effective cell representations as evaluated by broad downstreams.
>
> &nbsp;
>
> > **W2: Add scANVI baseline for the fine-tuning setting of cell type identification**
>
> Table A: Extended results for scANVI for cell type identification under fine-tuning setting. The classification (Clf.) and clustering (Clst.) performances on the ID dataset $D^{id}$ are reported.
>
> | **Method** | **Clf. Acc** | **Clf. Macro F1** | **Clst. AvgBio** |
> |--- |---  |--- |--- |
> | scCello | 0.867 | 0.511 | 0.694|
> | scANVI |0.382 | 0.024 | 0.472|
>
> &nbsp;
>
> As suggested, we add **scANVI** as baselines, implemented with scvi library [a]. As shown in Table A above, **scCello outperforms scANVI.**
>
> [a] Gayoso, et al. "A Python library for probabilistic analysis of single-cell omics data." Nature biotechnology 40.2 (2022): 163-166.
>
> &nbsp;
>
> > **W3: Whether the impressive batch correction results can extend from 10x assay to different sequencing technologies?**
>
> In principle, yes. Because scCello follows Geneformer’s **Rank Value Encoding (see App. C) to tokenize scRNA-seq data** into token sequences. As explained in Geneformer, this rank-based approach is **more robust against technical artifacts** than using raw numerical expressions, which can vary significantly across different assays.
>
> We acknowledge that generalizability to heterogeneous platforms is an important aspect of a TFM and may require additional consideration to address technical biases from the various sequencing protocols. However, we believe that our cell-ontology-informed TFM would be less affected by batch effect. We leave this as future work.
>
> &nbsp;
>
> > **Q1: The regularization loss (Eq. 2) could cause class collapse**
>
> Thanks for the constructive comment! We would like to explain that the regularization term would not lead to collapse empirically, primarily due to the masked gene prediction loss. This loss relies solely on the gene expression patterns in cells, making it hard for all cell representations $z_i$ to collapse into the linear transformation of their cell type representations $\textrm{Linear}(h_{c_i})$. In addition, the linear layer and the cell type representations were not fixed and were updated every batch
>
> We notice that this empirical justification was not mentioned in our paper, where our intuition for designing the regularization loss was to constrain the freedom of the optimization space.
> In our next version, we’ll add this in Sec. 2.3 for better explanation to avoid confusions.
>
> &nbsp;
>
> > **Q2: Which dataset did we use to train scVI for Tab. 1?**
>
> For all datasets in Tab. 1, including one ID and six OOD datasets, **scVI was trained on each of them individually.** We didn’t incorporate scCello’s pre-training dataset, because (1) scVI is not a foundation model and lacks a “pre-training stage”, and (2) even if we pre-train scVI, it does not have the capacity (in terms of both architectural design and parameter size) to be benefit from the pre-training on 22 millions cells.
>
> &nbsp;
>
> > **Q3: Whether the cell representation changes differ when knocking out housekeeping genes and marker genes?**
>
>
> For a case study, we identified six common housekeeping genes from literature [b]: ACTB, GAPDH, HPRT1, SDHA, UBC, and YWHAZ, and used known marker genes for “B cell” from CellMarker2 and PanglaoDB.
>
> We follow our approach in Sec. 4.4 and App. D.4 to perturb genes in-silico. For each cell of “B cell”, we calculate cell representations for (1) cell with no genes perturbed (2) cell with each of the six housekeeping genes perturbed (3) and cell with each of the B cell marker genes perturbed. To mitigate sample variance, we averaged these representations across all B cells for each gene. We visualize them via tSNE to compare the effects of perturbing housekeeping versus marker genes.
>
>
> **The tSNE figures are in Figure R3 in the PDF file attached to the global rebuttal response.** As shown in Figure R3, **knocking out housekeeping genes and marker genes leads to similar distribution shifts in cell representations.**
>
> As discussed in Sec. 5 and App. D.4, housekeeping genes could be predicted as positives. This is an issue for all TFMs in our evaluation, which does not influence model benchmarking. Given that housekeeping genes are well-documented and relatively few (\~400) compared to the extensive gene vocabulary (M = 25,424) in scCello, we can exclude them using external reference in our downstream pipeline. We leave this as future work.
> Additionally, knocking out marker genes is unlikely to shift cells to a different cell type unless two cell types differ by only one gene. In such cases, masking that gene could cause a shift to the other cell type.
>
> [b] Silver et al. "Selection of housekeeping genes for gene expression studies in the adult rat submandibular gland under normal, inflamed, atrophic and regenerative states." BMC molecular biology 9 (2008): 1-15.

---

> > ### Comment · Reviewer_BhVy · 2024-08-12
> >
> > I appreciate the authors' response to my review, and I thank them for the additional analyses they performed.
> >
> > I am surprised by the low performance of scANVI. Is that perhaps because of the bug documented and fixed here: *https://docs.scvi-tools.org/en/stable/tutorials/notebooks/scrna/scanvi_fix.html*? That being said, this is a recent bug fix, and the authors should not be penalized for using the older version of scANVI (if they even are).
> >
> > Lastly, I'd like to contest the claim that scVI cannot benefit from a pre-training dataset. This is the entire point of the scArches framework: *https://www.nature.com/articles/s41587-021-01001-7*.
> >
> > Overall still, I appreciate the performance contributions and extensive analyses the authors performed. I'd like to keep my score of 7.

---

> > > ### Author Response · Authors · 2024-08-14
> > >
> > > Dear Reviewer,
> > >
> > > Thanks for your great recognition and constructive comment for our project!
> > >
> > > For the scANVI performance, sorry that we were not aware about this bug fix. We checked the blog and double checked our implementation. It turns out that we used scvi==0.19.0 before the 1.1.0 version bug fix. We'll update the scANVI performance in our next version as additional baselines for the fine-tuning setting for cell type identification task.
> > >
> > > Also, thanks for bringing up and emphasizing the interesting idea that scVI can benefit from pre-training along with a referenced paper. We carefully read it and now understand better. We agree with this idea now, and will try to see if we can update the results for scVI with pre-training as well, if the computational time is affordable on 22 millions of cells.
> > >
> > > Thanks again for your time and efforts for reviewing our work!
> > >
> > > Best,
> > >
> > > Authors

---

### Official Review · Reviewer_yzGS · 2024-07-15

**Soundness:** 3
**Presentation:** 3
**Contribution:** 2
**Rating:** 6
**Confidence:** 4

**Summary:**

This paper introduces scCello, a new transcriptome foundation model (TFM), which learns cell representations over RNA gene expressions. Apart from using Masked language modeling (MLM), masking random gene expressions in cells, the work leverages structural knowledge from ontologies to improve the learned representations. It enforces cell type coherence, where cells of the same type should be close in representation space. Similarly, the representation of structurally close cells, measured over a PageRank measure, should be close. These structurally inferred objectives are encoded as separate contrastive losses. The pretaining objective tries to minimise the sum over the MLM loss and the two contrastive losses together with a regularisation term, which tackles class collapse. scCello is backed by a transformer-based encoder-only model with roughly 10.5 min parameters, which is rather small compared to competitors. The evaluation covers multiple downstream tasks and an ablation study over the optimisation objectives. The results show that scCello is either competitive with state of the art performance or improves it and that the aggregation of the losses is beneficial.

**Strengths:**

- Data: It's great that the work leverages diverse data sources (structural and token-based) and tries to incorporate the strengths of both of them by using custom losses.
- Experiments: The evaluation is extensive comparing the performance of the proposed model with multiple competitors and across multiple tasks. I also appreciated the ablation study and the analysis of overall performance wrt. #parameters.
- Model: With a reasonable size the produced model can surpass many larger-sized competitors.

**Weaknesses:**

**Incorporating structural knowledge**

GNN have been used for incorporating structural knowledge into pretraining of transformer-based models. For instance, take a look at GraphFormers (https://arxiv.org/abs/2105.02605). Such methods can be more elegant than the proposed PPR metric and the contrastive loss, since everything can be captured inside the model and no custom metric is necessary. While the author show that their proposed approach is beneficial for overall performance, such methods should be discussed or compared to, since they overlap in their goal of fusing structural and textual data with a major contribution of the paper.

**Limited Novelty**
The paper does not advance any method or approach. It is an application of existing methods for creating a TFM model.

**Questions:**

While you analysed the impact of the single training objectives, you didn't present numbers on different model sizes for scCello. Do you have any evidence how larger or smaller model impact downstream performances?

If ontologies and knowledge base grow, how does the training approach scale with them? Since you use contrastive loss, you could run into bottlenecks, if the number of negatives grows?

**Limitations:**

yes

---

> ### Author Rebuttal · Authors · 2024-08-06
>
> Thanks for your valuable comments! We respond to your concerns as below:
>
> &nbsp;
> > **W1: GNNs could offer a more elegant way to fuse structural knowledge into transformers than PPR metrics and contrastive objectives from scCello. GNN-fusion methods should be included as baselines.**
>
> We would like to argue that the **cellular ontology graph has unique challenges to be directly modeled by GNNs effectively**, supported by following evidence:
> 1. The ontology graph is **extremely sparse** with \~2.7k nodes and \~3.9k edges, and **faces long-distance issues**. For example, the average pairwise distance of \~400 nodes (i.e., cell types) associated with our pre-training scRNA-seq data is 7.39, and the maximum distance is 18.
> 2. Therefore, **it requires many layers (~7) of graph propagations of GNNs**, which may suffer from **over-smoothing** problems[a] for cell type representations. Even using a method like GraphFormer [b], which has only one layer of propagation in-between the transformers, the requirement of many layers of propagation leads to many layers of transformers. This would be **unaffordable given our computation resources**.
>
> We also would like to emphasize our claim on scCello’s contribution: previous TFMs ignored cell type lineage relationships between cells and treated each cell as independent training samples. To tackle this, scCello proposes to fuse ontology structure priors into TFM pre-training. **We acknowledge that a careful design of the GNN fusion method may further improve performance.**  We would like to leave it for future work.
>
> [a] Rusch, et al. "A survey on over-smoothing in graph neural networks." arXiv preprint arXiv:2303.10993 (2023).
>
> [b] Yang, et al. "Graphformers: Gnn-nested transformers for representation learning on textual graph." Advances in Neural Information Processing Systems 34 (2021): 28798-28810.
>
> &nbsp;
> > **W2: The paper lacks novelty because it is an application of existing methods for creating a TFM model.**
>
> Previous TFMs do not consider cell type lineage relationships between cells. **The key novelty of scCello is incorporating the cellular ontology graph into TFM pre-training** for better cell representation learning. **Incorporating an ontology graph is a non-trivial effort, since the ontology graph is sparse and faces long-distance issues** (see answer for W1). And in our method, the PPR metric and two inter- and intra- cellular contrastive objectives are simple and effective.
>
>
> scCello enjoys other novelties like:
> - **Simple but effective way to use cell type information**: Previous methods have proposed using cell type labels before, but treating each cell type independently with cell type classification loss has shortcomings. scCello uses pre-existing ontology graphs to determine the similarity between cell types, and this seems intuitively effective and empirically turns out to be true.
> - **New comprehensive evaluation for cell type clustering**: scCello evaluated not just on heldout in-distribution data, but also on unobserved cell types, tissues and donors.
> - **Novel algorithm for novel cell type classification**: scCello compares the similarity vector between a cell and prototype representations of each cell type to the similarity vector between a cell type and all other cell types, as derived from the ontology graph.
>
> &nbsp;
> > **Q1: How does the scaling of model sizes influence downstream performance?**
>
> **As discussed in Sec. 5, we leave scaling study as future work** because we lack the computational resources and scaling study is also not our focus in this paper. We focus on developing new algorithms for improving TFM pre-training strategy aiding various cell-related and gene-related downstream tasks.
>
> &nbsp;
> > **Q2: What’s the scaling complexity for training scCello with respect to the size of the ontology graph? When the number of negatives increases, the contrastive term in scCello faces a time bottleneck.**
>
> **The time complexity** of scCello includes that for PPR pre-calculation and TFM pre-training:
>
> - The pre-calculation needs $O(N*(I * (N + E)))$, where $N$ is the number of graph nodes, $E$ is the number of graph edges, and $I$ is the number of interactions. Specifically, for each target node, numerous iterations of graph message passing were run till convergence. Empirically, it was fast to accomplish and took less than several minutes to finish the ontology graph with 2.7k nodes. Since the Cell Ontology updates stay roughly on the scale of thousands of nodes [c], PPR calculation would still be fast in the coming future.
>
> - The TFM pre-training contains pairwise calculation for cells and their associated cell types within the batch, for the intra- and inter- cellular contrastive objectives. Its time complexity per batch for contrastive terms is roughly $O(B^2 * D)$, instead of $O(N^2 * D)$, where $B$ is the fixed batch size per GPU ($B=12$) and $D$ is the feature dimension. Therefore, the complexity is independent from N.
>
> **Therefore, the time complexity of scCello does not increase with the ontology graph size**.
>
>
> **The number of negatives** in scCello is fixed and equals the batch size minus 1 (i.e., $B-1$), following the common setting in contrastive learning papers [d,e,f]. Because the batch size $B$ in scCello is already maximized to fully utilize GPU memories, the number of negatives **cannot increase**.
>
> [c] Diehl, et al. "The Cell Ontology 2016: enhanced content, modularization, and ontology interoperability." Journal of biomedical semantics 7 (2016): 1-10.
>
> [d] Chen, et al. "A simple framework for contrastive learning of visual representations." International conference on machine learning. PMLR, 2020.
>
> [e] He, et al. "Momentum contrast for unsupervised visual representation learning." Proceedings of the IEEE/CVF conference on computer vision and pattern recognition. 2020.
>
> [f] Jaiswal, Ashish, et al. "A survey on contrastive self-supervised learning." Technologies 9.1 (2020): 2.

---

> > ### Comment · Reviewer_yzGS · 2024-08-12
> >
> > Thanks for addressing my comments and questions. Even though I only partially accept the affordability argument, since  affordability is not part of the paper's narrative. However, I agree that the model scaling experiments are not inherently important for the paper's focus. Further, I would encourage the authors to include a paragraph about GNN's in related work for camera ready.
> >
> > In summary, I increase my score from 5 to 6.

---

> ### Author Response · Authors · 2024-08-13
>
> Dear Reviewer,
>
> Thanks so much for your support on our project! In the final version for camera ready, we will include the discussion for GNNs according to the great suggestions.
>
> Best,
>
> Authors

---

### Author Rebuttal · Authors · 2024-08-06

We would like to appreciate all reviewers for your constructive suggestions and valuable comments on our paper!

Here is a brief summary of important points from all reviewers:

- **Performance comparison with more non-TFM traditional methods specialized for downstreams (Review BhVy, bUQe, E5ju)**: for the fine-tuning setting of cell type identification, we evaluate two more baselines scANVI and L1 Logistic Regression (L1-LR), and scCello outperforms both of them. For the marker gene prediction task, we incorporate Differential Expression Tests (DET) for comparison and scCello also outperforms it.

We also would like to highlight our contributions:

- **Innovation in integrating cell ontology graphs as priors**: scCello enhances the model’s understanding of biologically important cell type lineage relationships between cells using cell ontology graphs, improving its generalization and transferability
- **Introduction of a multi-level objective function**: scCello employs a simple yet effective multi-level objective pre-training strategy. It encompasses gene-level prediction, intra-cellular ontology coherence, and inter-cellular relational alignment, enabling the model to learn complex relationships between genes, cells, and cell types. This leads to more precise and robust cell representations for scCello.
- **Model effectiveness and comprehensive evaluation**: We conducted  a comprehensive set of experiments, and evaluated scCello’s effectiveness and superiority against various non-TFM and TFM baselines across broad downstream tasks. scCello is also robust to batch effects and parameter efficient, highlighting its strong potential to be applied in real-world applications.

**More detailed per-question feedback is presented below**. We use prefixes **“W”, “Q”, and “L” to indicate Weakness, Question, and Limitation** raised in reviewers’ comments.  We hope these feedbacks can address your concerns, and we welcome any further questions.

---

### Decision · Program_Chairs · 2024-09-25

**Decision:**

Accept (spotlight)

**Comment:**

This paper presents an algorithm, scCello, designed to enhance the pre-training of Transcriptome Foundation Models (TFM) by incorporating taxonomic relationships between cell types within cell ontology graphs. This approach aims to improve performance in cell-related and gene-related downstream tasks. Extensive experiments have been conducted to evaluate the proposed scCello in comparison with other TFM models across several real-world biological and drug response tasks.

All reviewers agree that the proposed method is practically useful, although some do not consider it novel theoretically. Nevertheless, all reviewers are convinced that the experiments are extensive and adequately support the narrative of the paper. Most of the questions raised concerned technical and experimental details, and reviewers were satisfied with the authors' responses; no reviewer lowered their rating, and one even raised theirs after the rebuttal.

This work is significant in applications and is worthy of being reported.